# Usp7 regulates Hippo pathway through deubiquitinating the transcriptional coactivator Yorkie

Xiaohan Sun[1], Yan Ding[1], Meixiao Zhan[2], Yan Li[1], Dongqing Gao[1], Guiping Wang[3], Yang Gao[3], Yong Li[2], Shian Wu[3], Ligong Lu[2], Qingxin Liu[1] & Zizhang Zhou[1]

The Hippo pathway plays an important role in organ development and adult tissue homeostasis, and its deregulation has been implicated in many cancers. The Hippo signaling relies on a core kinase cascade culminating in phosphorylation of the transcription coactivator Yorkie (Yki). Although Yki is the key effector of Hippo pathway, the regulation of its protein stability is still unclear. Here, we show that Hippo pathway attenuates the binding of a ubiquitin-specific protease Usp7 to Yki, which regulates Hippo signaling through deubiquitinating Yki. Furthermore, the mammalian homolog of Usp7, HAUSP plays a conserved role in regulating Hippo pathway by modulating Yap ubiquitination and degradation. Finally, we find that the expression of HAUSP is positively correlated with that of Yap, both showing upregulated levels in clinical hepatocellular carcinoma (HCC) specimens. In summary, our findings demonstrate that Yki/Yap is stabilized by Usp7/HAUSP, and provide HAUSP as a potential therapeutic target for HCC.

[1] State Key Laboratory of Crop Biology, College of Life Sciences, Shandong Agricultural University, 271018 Tai'an, China. [2] Center of Intervention radiology, Zhuhai Precision Medicine Center, Zhuhai People's Hospital, 519000 Zhuhai, China. [3] State Key Laboratory of Medicinal Chemical Biology, College of Life Sciences, Nankai University, 300071 Tianjin, China. These authors contributed equally: Xiaohan Sun, Yan Ding, Meixiao Zhan. Correspondence and requests for materials should be addressed to L.L. (email: luligong1969@126.com) or to Q.L. (email: liuqingxin@sdau.edu.cn) or to Z.Z. (email: zhouzz@sdau.edu.cn)

The Hippo (Hpo) pathway is known to play essential roles in modulating cell proliferation and apoptosis, thus contributing to organ size control, development, and tumorigenesis[1]. The Hpo pathway was initially discovered in *Drosophila* for its critical roles in restricting cell growth and promoting cell death[2–4]. The previous data have clearly demonstrated that Hpo pathway comprises several tumor-suppressor proteins, which form a core kinase cascade[5]. In *Drosophila*, the Ste20-like kinase Hpo, facilitated by a WW-domain-containing protein Salvador (Sav), phosphorylates and activates the kinase Warts (Wts), which then phosphorylates the downstream target protein Yorkie (Yki) with the assistance of its adaptor Mats[6]. Yki is a transcriptional coactivator that governs the expression of target genes through associating with the transcription factor Scalloped (Sd) in the nucleus[7]. Well-known Yki-Sd target genes include *diap1* and *cyclin E*, which respectively hampers apoptosis and promotes cell proliferation[2]. Therefore, inactivation of Hpo or Wts, or ectopic expression of Yki, results in substantial tissue overgrowth featured by excessive cell proliferation and decreased apoptosis[2,8]. The critical mechanism in the Hpo signaling transduction is regulation of Yki protein subcellular localization by phosphorylating Yki[2,8,9]. Activation of Hpo pathway results in Hpo/Wts-mediated Yki phosphorylation. 14-3-3 proteins bind the phosphorylated Yki in the cytoplasm and prevent its nuclear accumulation, culminating in suppressing Yki target gene expression[10]. Besides phosphorylation, whether Yki protein undergoes other modifications is still unclear. On the other hand, how the nuclear Yki terminates its signal is also a long-standing puzzle.

The Hpo pathway is evolutionarily conserved from *Drosophila* to mammals, as the core components and regulatory mechanisms are similar with few exceptions. In mammals, MST1/2 (Hpo orthologs) and the adaptor SAV1 (Sav ortholog) phosphorylates and activates the downstream kinase LATS1/2 (Wts orthologs). Then, LATS1/2 forms a complex with MOB1A/B (Mats orthologs) to phosphorylate the co-transcription factor Yap/TAZ (Yki orthologs)[9,11]. Yap functions together with TEAD1/2/3/4 (Sd orthologs) in the nucleus to turn on the transcription of target genes. Analogous to the case in *Drosophila*, Hpo pathway in mammals leads Last1/2-dependent Yap/TAZ phosphorylation, which results in Yap/TAZ cytoplasmic retention and inactivation through interacting with 14-3-3 proteins[12]. Otherwise, when Hpo pathway is off, Yap and TAZ enter into the nucleus to trigger the expression of pro-proliferative and anti-apoptotic genes[13]. Therefore, Yap has been shown to be critical for cancer initiation and growth[14,15].

Ubiquitination is one of the well-studied posttranslational protein modifications (PTMs) that plays important roles in many cellular processes, such as embryonic development, cell cycle, tumorigenesis, and signaling transduction[16,17]. The main outcome of ubiquitination modification is promoting protein degradation[17]. Many key components of Hpo pathway are revealed to be degraded by ubiquitination, including Yap[18], Lats1[19,20], Lats2[21,22], and Expanded (Ex)[23,24]. Ubiquitin-mediated protein modification is a reversible process due to the function of deubiquitinating enzymes (DUBs). Based on the mechanism of catalysis, previous studies have classified DUBs into six families[25,26]. Recently, a seventh family DUBs (ZUFSP family) has been discovered[27–29]. Usp7, also known as herpes virus-associated ubiquitin-specific protease (HAUSP), belongs to the USPs subfamily[30]. Increasing substrates of Usp7 have been identified, such as P53[31], Ci/Gli[32], and N-Myc[33]. Through deubiquitinating a wide range of targets, Usp7 regulates many physiological and pathological processes, including embryonic development and tumorigenesis[34,35].

Although recent studies have clearly demonstrated that various E3 ligases are involved in regulating the Hpo pathway, few DUBs have been implicated. In this study, via both loss-of-function and gain-of-function analyses, we identify that Usp7 regulates Hpo pathway through its deubiquitinase activity. We provide evidence that Usp7 binds Yki to counteract Yki ubiquitination and increase Yki protein level. HAUSP, the mammalian ortholog of Usp7, can functionally substitute Usp7 to modulate Hpo signaling in *Drosophila*. Furthermore, HAUSP also regulates Yap ubiquitination and Hpo pathway activity in mammalian systems. Finally, we found that the expression of HAUSP is upregulated in clinical hepatocellular carcinoma (HCC) specimens, positively correlated with Yap, providing HAUSP as a potential target for HCC treatment.

## Results

**Usp7 promotes the expression of Hpo target genes.** To explore the DUB that regulates Hpo-Yki pathway, we carried out a RNA interference (RNAi)-mediated screen, in which RNAi lines targeting DUBs were expressed in wings. Knockdown of *usp7* resulted in a small wing (Supplementary Fig. 1a), phenocopying activation of Hpo pathway. First of all, we generated mouse anti-Usp7 antibody and found that knockdown of *usp7* apparently decreased Usp7 signals, whereas overexpression of *usp7* elevated Usp7 signals (Figs. 1a, b), indicating that this antibody can specifically recognize Usp7 protein. Meanwhile, we found that *usp7* evenly expressed in the wing and eye discs (Supplementary Fig. 1b), and Usp7 protein mainly localized in the nucleus (Supplementary Fig. 1c). To investigate whether Usp7 modulates Hpo pathway, we silenced *usp7* in wing discs and checked the expression of Hpo pathway target genes. Knockdown of *usp7* apparently decreased *diap1*-lacZ expression (Figs. 1c, d, q). In addition, we also examined other well-known target genes. Compared with control discs (Figs. 1e, g), knockdown of *usp7* attenuated the expression of CycE and *ban*-lacZ (Figs. 1f, h, q). Furthermore, knockdown of *usp7* in the eye disc also decreased CycE level (Supplementary Fig. 1d). We also employed another *usp7* RNAi line, which targets distinct region of *usp7* gene, to validate this result (Supplementary Fig. 1e). Since previous reports have demonstrated that some interactions exist between Hpo and Hh pathway[36,37], we should test whether Usp7 regulates Hpo signaling activity through Hh pathway. In the wing disc, *ci* only expresses in the anterior (A) compartment, whereas *hh* exclusively expresses in the posterior (P) compartment[38]. Knockdown of *usp7* in the wing disc via *ApG4*, which expresses across the A and P compartments, equally decreased A- and P-compartmental *diap1*-lacZ levels (Supplementary Fig. 1f), removing the possibility that Usp7 modulates Hpo pathway through Hh signaling. Overall, these results suggest that Usp7 is involved in Hpo-Yki signaling regulation.

To avoid the off-target effect of RNAi, further experiments were performed using a null allele of *usp7*, *usp7^{KG06814}* (Supplementary Fig. 1g)[39]. Due to *usp7^{KG06814}* homozygote was embryonic lethal, we employed Flp recombinase/Flp recombinase target (FLP/FRT) technique to generate *usp7^{KG06814}* mutant clones in wing and eye discs and examined Yki target gene expression. *usp7^{KG06814}* clones, marked by the loss of green fluorescent protein (GFP) signals, showed decreased *Drosophila* inhibitor of apoptosis 1 (DIAP1) and *ban*-lacZ staining in wing discs (Figs. 1i, j). We also found that CycE protein was decreased in *usp7^{KG06814}* clones in the eye disc (Fig. 1k). Interestingly, we found that the areas of *usp7^{KG06814}* mutant clones were smaller than those of the neighbor twin spots, indicating that loss of *usp7* possibly hampers tissue growth. To validate this result, we generated some large *usp7^{KG06814}* clones and analyzed the area ratios of clones/twin spots. Compared with control clones,

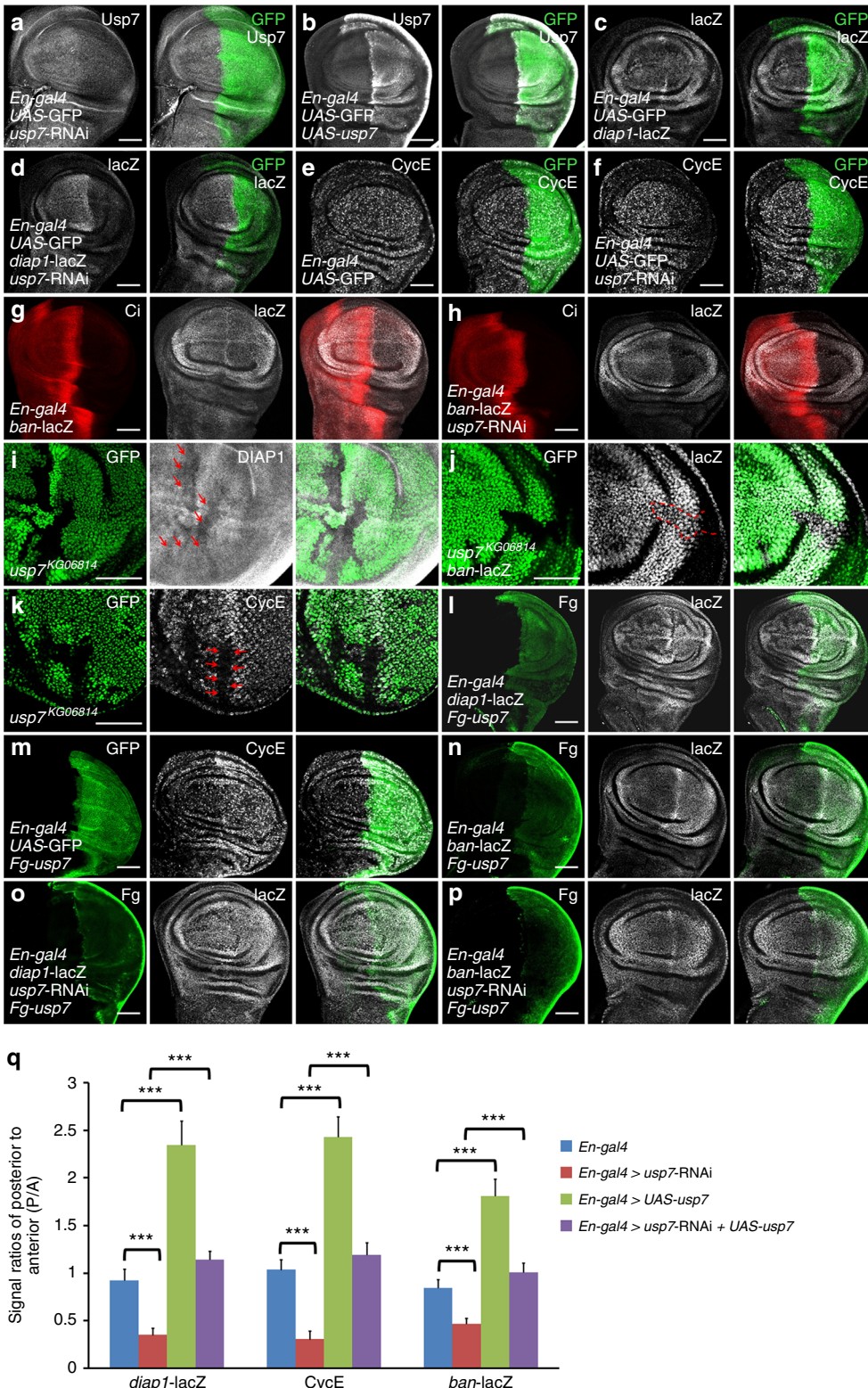

$usp7^{KG06814}$ clones showed apparent growth defect both in wing discs and in eye discs (Supplementary Fig. 1h).

Given that loss of $usp7$ attenuated Yki target gene expression, we next wanted to examine whether ectopic expression of $usp7$ could turn on these target genes. Overexpression of $usp7$ substantially increased the levels of $diap1$-lacZ (Figs. 1l, q), CycE (Figs.1m, q), and $ban$-lacZ (Figs. 1n, q). Furthermore, we found that overexpression of exogenous Fg-tagged $usp7$ could effectively restore the decreased $diap1$-lacZ (Figs. 1o, q), $ban$-lacZ (Figs. 1p, q), and small wing caused by $usp7$ RNAi (Supplementary Fig. 1a). On the other hand, we found that $usp7$ steadily expressed in different development stages from egg to adult (Supplementary Fig. 1i), and hyperactivated Yki did not affect $usp7$ expression (Supplementary Fig. 1j), indicating that the expression of $usp7$ is independent of Hpo-Yki pathway. Taken together, these findings suggest that Usp7 is a *bona fide* and constitutive regulator for Hpo-Yki signaling transduction.

**Fig. 1** Usp7 promotes the expression of Yorkie (Yki) target genes. All imaginal discs shown in this study were oriented with anterior to the left and ventral up. **a, b** Late third-instar wing discs of *usp7* knockdown (**a**) or *usp7* overexpression (**b**) were immunostained to show Usp7 (white) and GFP (green). GFP (green) marks the expression pattern of En-gal4 in the wing disc. Of note, mouse anti-Usp7 antibody could recognize Usp7 protein. **c-f** Wing discs of control (**c, e**) or expressing *usp7* RNA interference (RNAi) by En-gal4 (**d, f**) were stained to show GFP (green) and *diap1*-lacZ (white in **c, d**) or CycE (white in **e, f**). **g, h** Wing discs of control (**g**) or expressing *usp7* RNAi by En-gal4 (**h**) were stained to show Ci (red) or *ban*-lacZ (white). Ci exclusively expresses in the anterior compartment cells. **i-k** Wing discs (**i, j**) or eye disc (**k**) carrying *usp7*[KG06814] clones were stained to show the expression of GFP (green) and DIAP1 (white in **i**), *ban*-lacZ (white in **j**) or CycE (white in **k**). *usp7*[KG06814] clones are recognized by the lack of GFP. Of note, *usp7* mutant cells exhibited decrease of DIAP1 (marked by arrows in **i**), *ban*-lacZ (marked by dash line in **j**), and CycE (marked by arrows in **k**). **l-n** Overexpression of *usp7* by En-gal4 increased *diap1*-lacZ (**l**), CycE (**m**) and *ban*-lacZ (**n**). **o, p** Wing discs simultaneous expressing *usp7* RNAi and Fg-*usp7* were stained to show Fg tag (green) and *diap1*-lacZ (white in **o**), or *ban*-lacZ (white in **p**). Of note, the decreased *diap1*-lacZ and *ban*-lacZ caused by *usp7* knockdown were restored by the expression of Fg-*usp7*. **q** Quantification analyses of indicated signals. Data are means ± SEM. *n* = 4 biological-independent discs. ***P < 0.001 by Student's *t*-test. Scale bars: 50 μm for all images

**Usp7 acts downstream of Wts, upstream of Yki**. Hpo pathway is defined as a kinase cascade whereby Hpo phosphorylates and activates Wts, in turn, Wts phosphorylates and inactivates the transcriptional cofactor Yki to suppress target gene expression[1,11]. Through both loss-of-function assays and gain-of-function assays, we have clearly demonstrated that Usp7 is involved in regulating Hpo pathway. We next wanted to ask whether Usp7 genetically interacts with any components of Hpo signaling. Knockdown of *hpo* increased CycE protein level (Fig. 2a), which was reversed by *usp7* RNAi (Fig. 2b). Additionally, another target *diap1*-lacZ showed the similar result (Figs. 2c, d), indicating that Usp7 sits downstream of Hpo to control Hpo pathway. Analogous to *hpo*, the elevated CycE and *diap1*-lacZ caused by *wts* RNAi were overcome by *usp7* knockdown (Figs. 2e–h). As Hpo pathway is a major regulator of cell cycle, we also monitored cell proliferation in eye discs. In eye discs, cells posterior to the morphogenetic furrow (MF) undergo a synchronous second mitotic wave (SMW) that can be labeled by Bromodeoxyuridine (BrdU) incorporation[40]. Few BrdU-positive cells are found posterior to the SMW. Knockdown of *usp7* decreased (Supplementary Fig. 2a, b), whereas overexpression of *usp7* increased BrdU-positive cells (Supplementary Fig. 2c) in UAS-*wts* background, suggesting that Usp7 acts downstream of Wts.

As Usp7 regulates Hpo signaling downstream of Hpo and Wts, it is worthy to test the relationship between Yki and Usp7. Knockdown of *yki* decreased *diap1*-lacZ expression (Fig. 2i), which failed to be restored by *usp7* overexpression (Fig. 2j). On the other hand, the upregulated CycE induced by *yki* overexpression could not be remitted by silencing *usp7* (Figs. 2k, l). We found that knockdown of *usp7* decreased Yki protein (Fig. 2m). Furthermore, loss of *usp7* also downregulated Yki level (Fig. 2n and Supplementary Fig. 2d), without affecting the transcription of *yki* (Supplementary Fig. 2e).

The Yki protein contains two WW domains, which are required for its transcriptional activity[41]. Overexpression of a low activity form, *yki-1w*, which only harbors one WW domain, increased *ban*-lacZ level (Fig. 2o). However, simultaneous overexpression of *usp7* and *yki-1w* further elevated *ban*-lacZ level and resulted in serious overgrowth (Fig. 2p). Furthermore, we employed flp-out method to generate GFP-positive clones. Compared with *usp7* or *yki-1w* overexpression alone (Supplementary Fig. 2f, g), simultaneous expressing *yki-1w* and *usp7* produced larger and rounder clones (Supplementary Fig. 2h). Collectively, these results indicate that Usp7 regulates Hpo pathway downstream of Hpo and Wts, upstream of Yki.

**Usp7 interacts with Yki**. Given that the deubiquitinase plays its roles always through binding and deubiquitinating substrates[42,43], we speculated that interaction with a component of Hpo pathway was essential for Usp7 controlling Hpo signaling. To test this possibility, co-immunoprecipitation (co-IP) experiments were performed in S2 cells. Consistent with above epistasis analysis, we found that Usp7 reciprocally bound Yki (Figs. 3c–f), but not other Hpo pathway components (Supplementary Fig. 3a–i), suggesting that Usp7 possibly regulates Hpo pathway through direct targeting Yki.

Usp7 contains a MATH domain in its N-terminus (Fig. 3a), which is responsible for binding some substrates, including Ci and P53[31,32]. Yki protein harbors two WW domains in its C-terminal region (Fig. 3b), which is important for its interaction with several partners, including Ex and Taiman[44,45]. Most regulating events in the Hpo pathway are mediated by WW-domain–PPxY interactions[46]. Intriguingly, a search in Usp7 protein revealed one PPxY degron ([121]PPV[124]Y) just located in the MATH domain. To test whether Yki binds Usp7 via WW-domain–PPVY interaction, we generated Usp7-ΔMATH truncated construct and performed co-IP experiment. The co-IP results showed that both Usp7 and Usp7-ΔMATH could pull-down Myc-Yki (Fig. 3g), indicating that Usp7 binds Yki in a MATH-independent manner. Furthermore, Yki did not affect the interaction between Usp7 and Ci, a substrate of Usp7 that binds its MATH domain (Fig. 3h). This is consistent with the fact that Usp7 binds Yki not via MATH domain. On the other hand, we revealed that the N-terminus, but not C-terminus of Yki was responsible for its interaction with Usp7 through both co-IP and glutathione-S-transferase (GST) pull-down assays (Figs. 3i–k). To map which region of Usp7 is responsible for Yki interaction, we generated various Usp7 truncated mutants. The co-IP result showed that Usp7 bound Yki-N through its C-terminal fragment (Supplementary Fig. 3j). In sum, our findings demonstrate that Usp7 binds Yki through a MATH-domain- and WW-domain-independent manner.

We next tested whether Hpo signaling affects Usp7-Yki interaction. Wild-type Hpo substantially blocked Usp7-Yki association, whereas a kinase dead form of Hpo (Hpo-KD) exerted an opposite effect (Fig. 3l). Consistently, the other kinase Wts played a similar role in regulating Usp7–Yki interaction (Fig. 3m). These results indicate that Hpo pathway likely decrease Yki protein by attenuating Usp7–Yki association. Consistently, overexpression of *hpo* or *wts* indeed decreased Yki protein level (Supplementary Fig. 3k). Contrarily, knockdown of *hpo* or *wts* elevated Yki, which was restored by *usp7* RNAi (Supplementary Fig. 3l).

**The deubiquitinase activity is essential for Usp7-regulating Hpo pathway**. Although above biochemical data showed that Usp7 binds Yki not through its MATH domain, we should confirm this result via in vivo assay. Overexpression of *usp7-ΔMATH*, in which the MATH domain was deleted, still elavated the expresssion of *diap1*-lacZ (Fig. 4a), *ban*-lacZ (Fig. 4b), and CycE (Supplementary Fig. 4b), supporting that the MATH domian is dispensable for Usp7-regulating Hpo pathway.

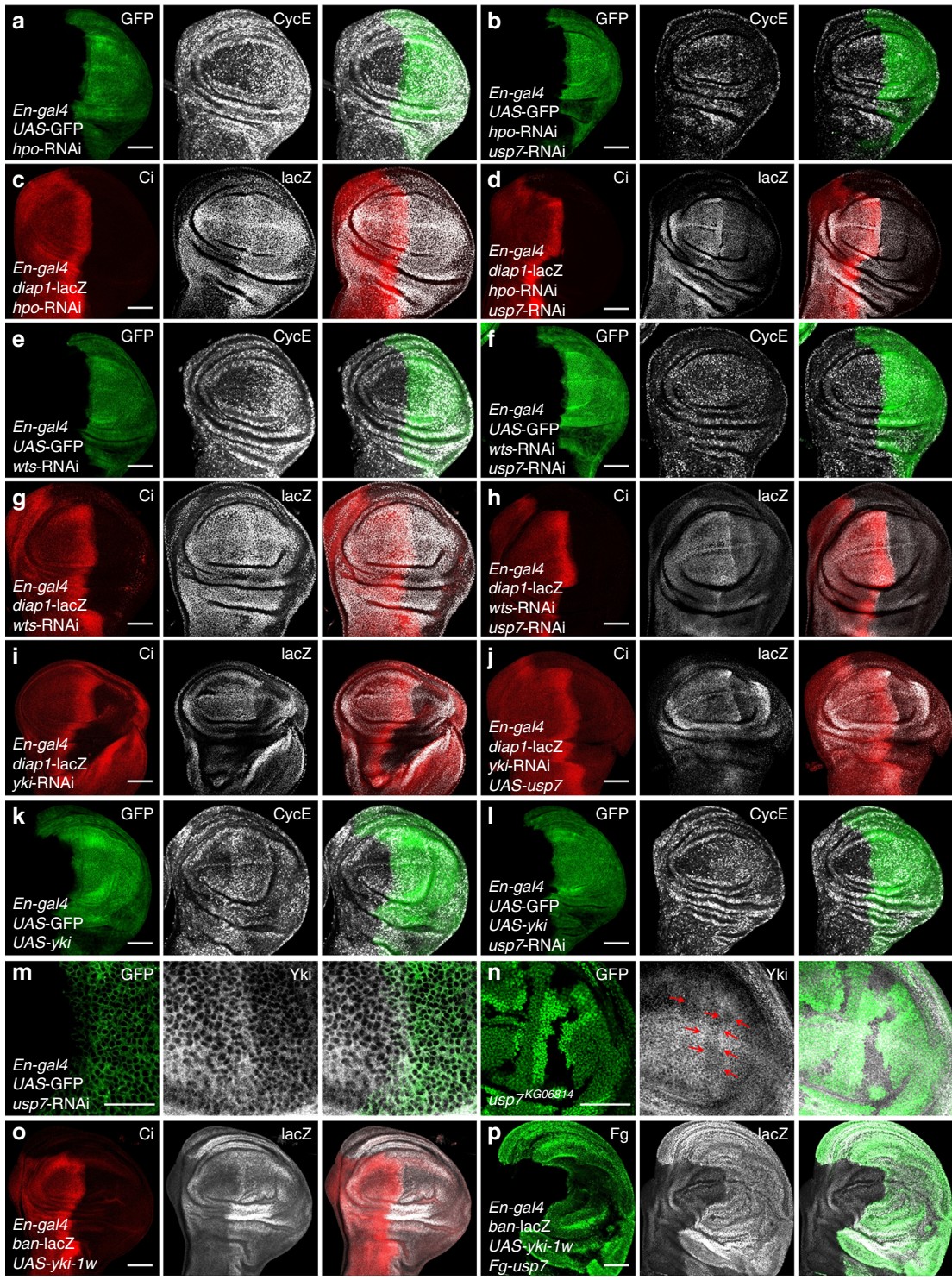

Each member of ubiquitin specific proteases (UBPs) family always contains a characteristic cysteine (Cys) site at the core catalytic domain[47]. Substitution of this Cys with alanine (Ala) often destroys its deubiquitinase capability. To examine the importance of Usp7 deubiquitinase activity for its regulation of Hpo pathway, we generated a deubiquitinase deficiency mutant of Usp7 with Cys250 replaced by Ala (*usp7-CA*). Co-IP result showed that this point mutation on Usp7 did not affect its interaction with Yki (Supplementary Fig. 4a). Overexpression of *usp7-CA* decreased the expression of *diap1*-lacZ (Fig. 4c), *ban*-lacZ (Fig. 4d), and CycE (Supplementary Fig. 4c), suggesting that Usp7-CA possibly plays a dominant negative role. In addition,

rescue experiments revealed that *usp7-CA* failed to restore the decrease of *ban*-lacZ caused by *usp7* knockdown (Fig. 4e). However, *usp7-ΔMATH* could rescue *ban*-lacZ level under *usp7* knockdown (Fig. 4f).

As the activity of Hpo pathway closely links eye size, we turned to examine whether Usp7 modulates eye size under Hpo-activated background. Compared with the control eye (*GMR-gal4* > UAS-*wts*), knockdown of *usp7* or overexpression of *usp7-CA* led to a smaller eye (Fig. 4g), whereas *usp7* or *usp7-ΔMATH* overexpression increased the eye size (Fig. 4g). In addition, we got the similar result under *yki* heterozygous mutant background (*yki*^B5/+) (Fig. 4h). Overall, Usp7 modulates Hpo pathway

**Fig. 2** Usp7 regulates Hippo (Hpo) pathway through Yorkie (Yki). **a** A wing disc with *hpo* knockdown was stained to show GFP (green) and CycE (white). **b** A wing disc simultaneous expressing *hpo* RNA interference (RNAi) and *usp7* RNAi was stained to show GFP (green) and CycE (white). Of note, knockdown of *usp7* decreased CycE level in *hpo* RNAi background. **c** A wing disc with *hpo* knockdown was stained to show Ci (red) and *diap1*-lacZ (white). **d** Knockdown of *usp7* attenuated *diap1*-lacZ expression in *hpo* RNAi background. **e** A wing disc expressing *wts* RNAi was stained to show GFP (green) and CycE (white). **f** Knockdown of *usp7* decreased CycE protein level under *wts* RNAi background. **g** A wing disc expressing *wts* RNAi was stained to show Ci (red) and *diap1*-lacZ (white). **h** Knockdown of *usp7* decreased *diap1*-lacZ expression under *wts* RNAi background. **i** Knockdown of *yki* decreased *diap1*-lacZ expression. **j** A wing disc simultaneous expressing *yki* RNAi and UAS-*usp7* was stained to show Ci (red) and *diap1*-lacZ (white). Of note, overexpression of *usp7* could not restore the decreased *diap1*-lacZ caused by *yki* knockdown. **k** A wing disc with *yki* ectopic expression by *En-gal4* was stained to show GFP (green) and CycE (white). Overexpression of *yki* resulted in CycE upregulation and overgrowth. **l** Knockdown of *usp7* did not rescue the phenotypes induced by *yki* overexpression. **m** Knockdown of *usp7* decreased Yki protein level. **n** A wing disc carrying *usp7*[KG06814] clones were stained to show the expression of GFP (green) and Yki (white). Of note, *usp7* mutation resulted in decreased Yki expression (marked by arrows). **o** A wing disc expressing *yki-1w* was stained to show Ci (red) and *ban*-lacZ (white). **p** A wing disc simultaneous expressing *yki-1w* and Fg-*usp7* was stained to show Fg (green) and *ban*-lacZ (white). Notably, overexpression of *usp7* further elevated *ban*-lacZ level and resulted in serious overgrowth. Scale bars: 50 μm for all images

activity in a deubiquitinase-dependent, but MATH-independent manner.

**Usp7 deubiquitinates Yki**. At present, not much is known about the homeostatic regulation of Yki protein. To address this issue, we carried out pulse-chase experiments in S2 cells and found that both exogenous Myc-Yki protein and endogenous Yki protein were unstable primarily due to lysosome-mediated protein turnover because lysosome inhibitor $NH_4Cl$ effectively blocked Yki protein degradation (Figs. 5a, b). Given that Yki protein mainly localized in the cytoplasm (Supplementary Fig. 4c) and entered into the nucleus when Hpo pathway was closed, we then asked whether the nuclear Yki and cytoplasmic Yki underwent degradation in the same manner. We separated the cytoplasmic and nuclear protein, and tested the degradation of Yki protein. Interestingly, the cytoplasmic Yki was destabilized via lysosome, whereas the nuclear Yki was degraded by proteasome (Fig. 5c). To further validate this result, we fused a nuclear localization sequence (NLS) on the C-terminus of Yki to generate a Myc-Yki-NLS variant, which was predominantly localized in the nucleus (Supplementary Fig. 4d), and found that it was destabilized mainly by proteasome (Fig. 5d and Supplementary Fig. 4j). On the other hand, we also generated a membrane-tethered form of Yki by adding a myristoylation signal at its N-terminus (Myc-Myr-Yki), which exclusively localized in the cytoplasm (Supplementary Fig. 4e), and found that it was degraded by lysosome (Fig. 5e and Supplementary Fig. 4k). Taken together, these results suggest that the cytoplasmic Yki protein undergoes lysosome-mediated turnover, whereas the nuclear Yki mainly undergoes proteasome-mediated proteolysis.

Usp7 specifically binds to Yki and its deubiquitinase activity is indispensable for Usp7-regulating Hpo pathway, implying that Usp7 may be required for the maintenance of Yki protein stability. In line with this, Usp7 indeed prevented Yki and Yki-NLS degradation (Fig. 5f and Supplementary Fig. 4l). In contrast, knockdown of *usp7* quickened Yki degradation (Fig. 5g). Given that Usp7 protein mainly localized in the nucleus, it is necessary to test whether Usp7 stabilizes nuclear Yki protein. Consistently, Usp7 increased the nuclear Yki abundance in a deubiquitinase activity-dependent manner, without obvious effect on the cytoplasmic Yki protein (Figs. 5h, i). Knockdown of *usp7* decreased nuclear Yki protein regardless of *hpo*-dsRNA/*wts*-dsRNA treatment (Fig. 5j), suggesting that Usp7 sits downstream of Hpo/Wts to stabilize nuclear Yki. Cell staining assays also showed that Usp7 promoted Yki nuclear accumulation (Supplementary Fig. 4f). As a matter of fact, nuclear accumulation of Yki protein upon Usp7 co-transfection is compatible with two possibilities. First, Usp7 tethers Yki protein in the nucleus through protein–protein interaction. Alternatively, Usp7 stabilizes the nuclear Yki via deubiquitinating Yki. Compared with control

cells (Supplementary Fig. 4c, g), both Usp7 and Usp7-ΔMATH promoted Yki nuclear accumulation (Supplementary Fig. 4f, i), whereas Usp7-CA had no effect (Supplementary Fig. 4h). These results showed that the deubiquitinase activity is essential for Usp7 promoting Yki nuclear accumulation, supporting the latter possibility.

We next sought to determine whether Usp7 deubiquitinates Yki using a cell-based ubiquitination assay[48]. We found that knockdown of *usp7* increased Yki ubiquitination (Fig. 5k). In contrast, both Usp7 and Usp7-ΔMATH decreased Yki ubiquitination, whereas Usp7-CA promoted Yki ubiquitination (Fig. 5l). Furthermore, Usp7 also decreased the ubiquitination levels of Yki-NLS, whereas Usp7-CA increased them (Fig. 5m). As Usp7 mainly localizes in the nucleus, it likely deubiquitinates nuclear Yki. We separated the cytoplasmic and nuclear protein for ubiquitination assay, and found that Usp7 robustly deubiquitinated nuclear Yki, not cytoplasmic Yki (Supplementary Fig. 5a). In line with this result, Usp7 effectively removed ubiquitins from Yki-NLS, not Myr-Yki (Supplementary Fig. 5b), supporting that Usp7 mainly deubiquitinates nuclear Yki. Hpo pathway governs Yki subcellular localization via Wts-mediated Yki phosphorylation on S168[9]. We generated phosphorylation-mimic (Yki-SD, Yki-S168D) and phosphorylation-deficient (Yki-SA, Yki-S168A) Yki mutants, and performed the ubiquitination assay. It showed that Usp7 robustly deubiquitinated Yki-SA, not Yki-SD (Supplementary Fig. 5c).

During protein ubiquitination, ubiquitin is always attached to the lysine residue of the substrate. To examine whether the ubiquitin modification of Yki occurs on lysine, we first replaced all 14 lysines by arginines to generate Myc-Yki-KallR mutant. Yki-KallR was stable and failed to be ubiquitination (Supplementary Fig. 5d, e), supporting that ubiquitins attach to lysine residues on Yki protein. To further figure out which lysines are responsible for ubiquitination, we carried out a prediction via an online web server (http://bdmpub.biocuckoo.org) and identified five potential ubiquitination sites (K52, K82, K93, K246, and K274). Compared with wild-type Yki, only mutation of K93 (K93R) significantly blocked Yki degradation and ubiquitination (Supplementary Fig. 5f–h), providing K93 of Yki is the ubiquitination site. Taken together, these findings suggest that the nuclear Yki protein undergoes ubiquitination-mediated turnover, which is antagonized by Usp7.

**Mammalian Usp7 is functionally conserved**. It has been demonstrated that the mammalian ortholog of Yki, Yap undergoes β-transducin repeat-containing protein (β-TrCP)-mediated ubiquitination and proteasome-dependent proteolysis[18]. To test whether the regulation of Usp7 on Yki is functionally conserved, we cloned *hausp*, the ortholog of *usp7* in human, into the UAS-Fg backbone vector to generate UAS-Fg-*hausp* transgenic flies. At

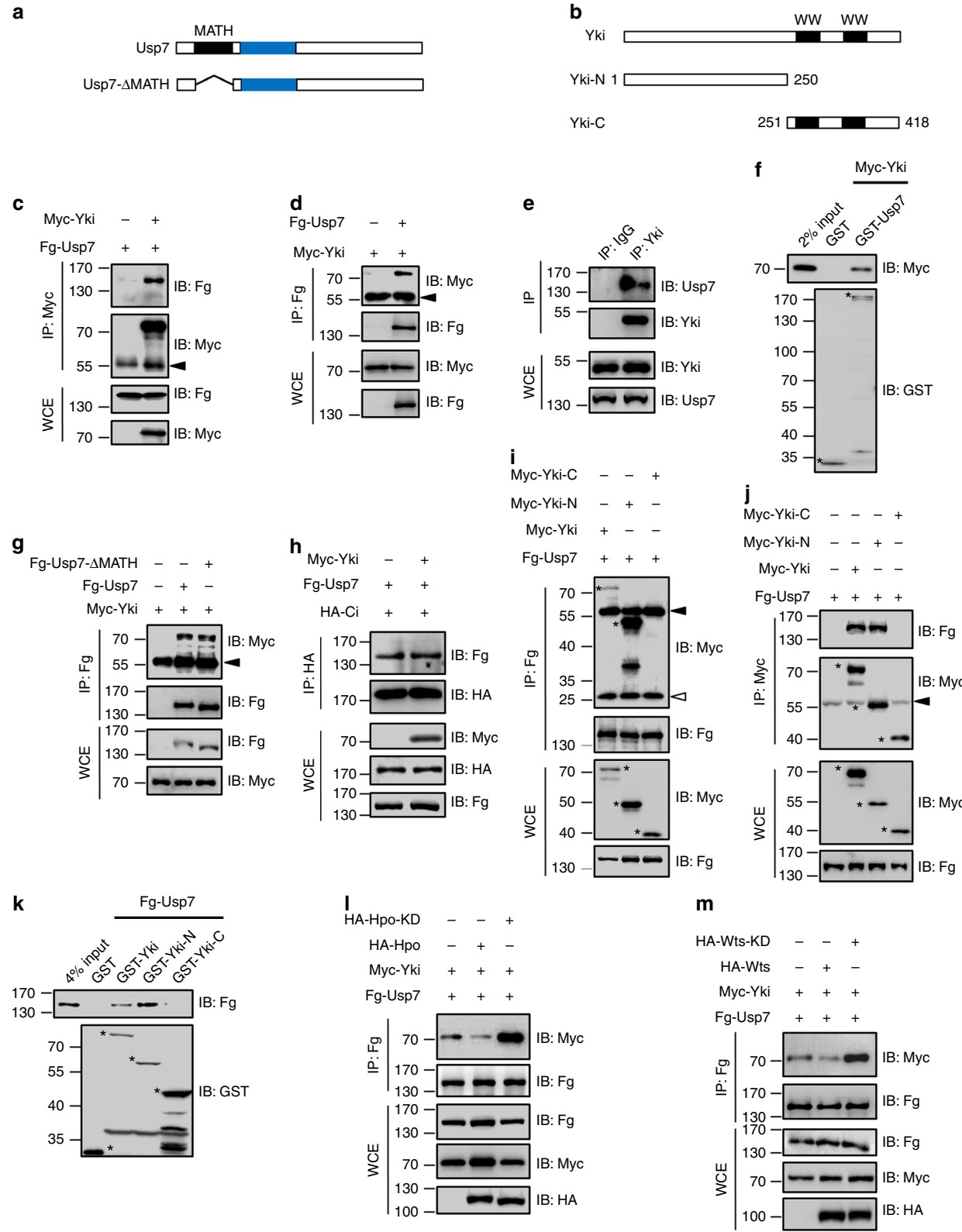

first, the co-IP assay showed that human HAUSP could pull-down *Drosophila* Yki (Fig. 6e). Misexpression of *hausp* by *En-gal4* upregulated *ban*-lacZ (Fig. 6a) and *diap1*-lacZ (Fig. 6b) levels, implying that HAUSP also serves as a positive regulator for Yki target genes. Furthermore, overexpression of *hausp* rescued the decreased *ban*-lacZ (Fig. 6c), *diap1*-lacZ (Fig. 6d), and small wing (Supplementary Fig. 1a) caused by *usp7* knockdown, suggesting that HAUSP could substitute *Drosophila* Usp7 to regulate Hpo

signaling. The mutant *usp7^{KG06814}* is a null allele, which the homozygote is lethal before the first instar larvae (Supplementary Fig. 6a). Overexpression of *hausp* using *actin*-gal4 could rescue the lethality of *usp7^{KG06814}* homozygotes (Supplementary Fig. 6a), further implying that HAUSP could functional replace *Drosophila* Usp7.

We next determined whether HAUSP modulates Yap stability and Hpo pathway activity in mammalian cells. The co-IP result

**Fig. 3** Usp7 binds Yorkie (Yki). **a, b** Schematic drawings show the domains or motifs in Usp7 (**a**) and Yki (**b**) and their truncated fragments used in subsequent co-IP and pull-down assays. Black and blue bars denote MATH domain and core catalytic domain of Usp7 (**a**). Black bars represent WW domains of Yki (**b**). **c** Immunoblots of immunoprecipitates (top two panels) or lysates (bottom two panels) from S2 cells expressing indicated constructs. Of note, Myc-Yki could pull-down Fg-Usp7 in S2 cells. **d** Fg-Usp7 could pull-down Myc-Yki in S2 cells. **e** Endogenous Yki protein pulled down endogenous Usp7 in S2 cells. **f** Extracts from S2 cells expressing Myc-Yki were incubated with GST or GST-Usp7. The bound proteins were analyzed by IB. Asterisks mark GST fusion proteins. **g** Usp7-ΔMATH could bind Yki. **h** Yki did not compete with Ci to bind Usp7. **i-j** Usp7 exclusively interacted with N-terminal region of Yki in S2 cells. Asterisks mark expressed Yki truncated fragments. **k** Pull-down between Fg-Usp7 and GST or GST-tagged Yki fragments. Asterisks indicate expressed GST fusion proteins. **l** Hippo (Hpo) decreased, whereas the kinase dead form Hpo-KD increased Usp7-Yki interaction. **m** Wts attenuated, whereas the kinase dead form Wts-KD promoted Usp7-Yki association. Above all, the solid arrowhead indicates heavy IgG (**c, d, g, i, j**), whereas the hollow arrowhead light IgG (**i**)

showed that exogenous Myc-Yap interacted with exogenous Fg-HAUSP in 293T cells (Fig. 6f). In addition, endogenous Yap could pull-down endogenous HAUSP and vice versa in 293T cells (Figs. 6g, h). The interaction of Yap and HAUSP was attenuated by LATS1/2 and MST1/2 (Figs. 6i, j). Overexpression of *hausp* substantially increased Yap protein level (Fig. 6k), whereas knockdown of *hausp* decreased Yap protein (Fig. 6l). Furthermore, we found that either knockdown or overexpression of the known HAUSP target genes (*p53, gli1, gli2, gli3,* and *N-Myc*) did not show any detectable effect on Yap protein level (Supplementary Fig. 6b–d), removing the possibility that HAUSP indirectly stabilizes Yap through these transcription factors. Consistent with the previous study[18], β-TrCP did degrade Yap in a dose-dependent manner, which was blocked by HAUSP, but not by HAUSP-CA (a deubiquitinase-deficiency form) (Fig. 6m).

We next sought to test whether HAUSP deubiquitinates Yap in mammalian cells. As expected, HAUSP indeed decreased Yap ubiquitination level, whereas HAUSP-CA exerted an opposite effect (Fig. 6n). In addition, we further revealed that HAUSP blocked β-TrCP-mediated Yap ubiquitination in a deubiquitinase-dependent manner (Figs. 6o, p). Intriguingly, the ubiquitination site (K93) of Yki was evolutionarily conserved between *Drosophila* and human (Supplementary Fig. 6e). Mutation of the corresponding lysine (K90) to arginine blocked Yap degradation (Supplementary Fig. 6f) and ubiquitination (Supplementary Fig. 6g). Meanwhile, we found that Yap-K90R showed higher efficiency in promoting cell proliferation than wild-type Yap (Supplementary Fig. 6h), together suggesting that K90 is a potential ubiquitination site of Yap protein.

The ability of HAUSP to antagonize Yap ubiquitination and degradation implies that HAUSP possibly affects Hpo pathway activity. To test this hypothesis, we carried out real-time quantitative polymerase chain reaction (RT-qPCR) experiments and revealed that knockdown of *hausp* decreased, whereas overexpression of *hausp* elevated the expression of Yap target genes, including *AREG, CTGF,* and *Cyr61* (Figs. 6q, r). The *CTGF*-luciferase reporter assay in 293T cells further confirmed this result (Fig. 6s). Moreover, neither knockdown nor over-expression of *hausp* affected *yap* mRNA levels (Figs. 6q, r), supporting that HAUSP regulates Yap through stabilizing Yap protein.

**HAUSP is upregulated in HCC tissues, and positively correlates with Yap**. Our above studies have demonstrated Usp7/HAUSP regulates Yki/Yap stability and Hpo pathway outputs in *Drosophila* and mammalian cells. To test whether HAUSP contributes tumorigenesis, we chose HCC specimens since liver is particularly sensitive to changes of Hpo signaling and deregulation of Hpo pathway closely links to liver cancers[9,49–51]. To address this question, we assessed HAUSP protein levels in 60 pairs of matched liver tissue samples. Consistent with the previous findings[52], HAUSP was upregulated in 49 of 60 HCC tumors compared with corresponding normal liver counterparts

(Fig. 7a and Supplementary Fig. 7a). We further checked the levels of Yap in these samples and revealed that Yap was higher (51 of 60) in HCC samples than that in matched nontumor tissues (Fig. 7b and Supplementary Fig. 7a). Furthermore, a strong positive correlation was found between HAUSP and Yap at the protein level (Fig. 7c). Consistently, RT-qPCR results showed that the expression of Yap target genes was indeed elevated in HCC samples (Fig. 7d).

To explore whether HAUSP is involved in regulating Hpo pathway and tumorigenesis in HCC, we employed a HCC cell line, Huh7. Transfection of HAUSP apparently increased mRNA levels of Yap target genes, where HAUSP-CA had no such effect (Fig. 7e). On the other hand, knockdown of *hausp* inhibited the expression of Yap target genes (Fig. 7f). In addition, we performed 3-(4,5-dimethylthiazol-2-yl)-2,5-diphenyltetrazolium bromide (MTT) experiments to analyze the proliferation of Huh7 cells after distinct treatments. Transfection of HAUSP promoted Huh7 cell proliferation, whereas HAUSP-CA did not show any effect (Fig. 7g). Knockdown of *hausp* hampered Huh7 cell proliferation, which was restored by *yap* overexpression (Fig. 7g), suggesting that HAUSP promotes proliferation of Huh7 cells through Yap. The expression of indicated constructs and the knockdown efficiency of indicated small interfering RNAs (siRNAs) were checked via immunoblotting (IB) (Supplementary Fig. 7b).

As a matter of fact, several inhibitors of HAUSP have been under preclinical experiments. P5091 triggers cell apoptosis in multiple myeloma cells and overcomes bortezomib resistance by inhibiting HAUSP activity[53]. In addition, P5091 treatment blocked medulloblastoma cell proliferation and metastasis through antagonizing HAUSP-mediated Gli stabilization[54]. We next wanted to test whether the inhibitor regulates Yap protein abundance and then the proliferation of HCC cells. Huh7 cells were treated with different concentrations of P5091 for 24 h, and then were harvested for western blot and RT-qPCR assays. P5091 decreased Yap protein in a concentration-dependent manner, but did not affect HAUSP protein (Fig. 7h). Consistently, P5091 treatment also downregulated the protein levels of Yap targets, including AREG and Cyr61 (Fig. 7h). We validated these results via RT-qPCR and found that P5091 indeed decreased Yap target genes expression, without affecting *yap* mRNA level (Fig. 7i). Furthermore, we carried out MTT experiments and found that P5091 attenuated Huh7 cell proliferation in a dose-dependent manner (Fig. 7j). We chose another HCC cell line to confirm this result, and found that P5091 also inhibited 7721 cell proliferation in a dose-related manner (Supplementary Fig. 7c). Since the previous studies have demonstrated that P5091 also inhibits Usp47 activity, we chose other specific Usp7 inhibitors (Usp7-IN-1[55] and HBX-19818[56]) to validate our results. Similar to P5091, Usp7-IN-1 and HBX-19818 decreased Yap protein in a dose-dependent manner, without affecting HAUSP levels (Fig. 7k). Furthermore, Both Usp7-IN-1 and HBX-19818 blocked HCC cell proliferation (Supplementary Fig. 7d). Consistently, these inhibitors could relieve the deubiquitinating effect of HAUSP upon Yap

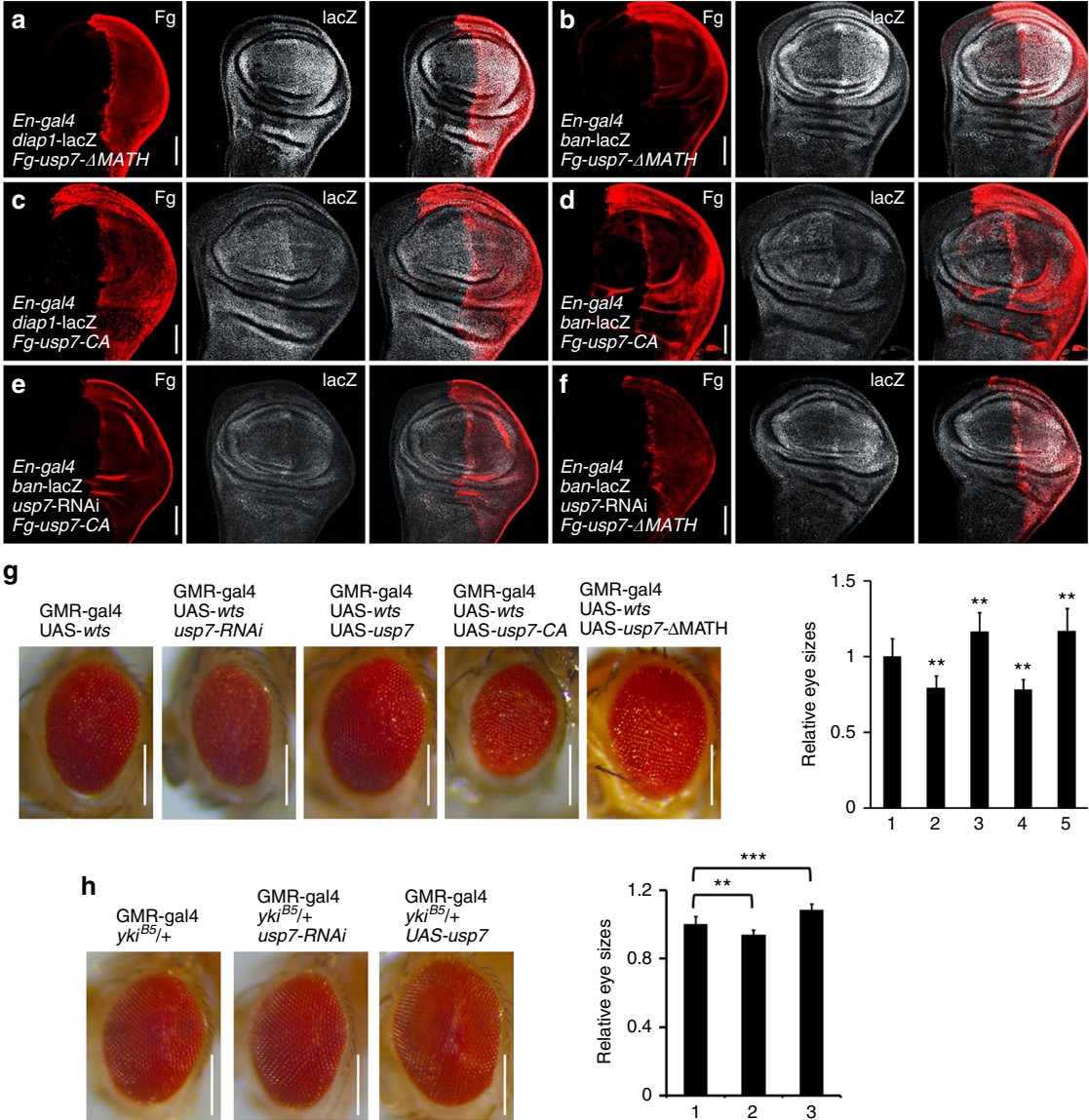

**Fig. 4** Usp7 regulates Hippo (Hpo) pathway though its deubiquitinase activity. **a, b** Wing discs expressing Fg-tagged *usp7-ΔMATH* were stained to show Fg tag (red) and *diap1*-lacZ (white in **a**) or *ban*-lacZ (white in **b**). Of note, overexpression of *usp7-ΔMATH* increased *diap1*-lacZ and *ban*-lacZ. **c, d** Wing discs expressing Fg-*usp7-CA* were stained to show Fg tag (red) and *diap1*-lacZ (white in **c**) or *ban*-lacZ (white in **d**). Of note, overexpression of *usp7-CA* decreased *diap1*-lacZ and *ban*-lacZ. **e** A wing disc simultaneous expressing *usp7* RNA interference (RNAi) and Fg-*usp7-CA* was immunostained to show Fg (red) and *ban*-lacZ (white). Overexpression of Fg-*usp7-CA* could not restored the decreased *ban*-lacZ caused by *usp7* knockdown. **f** Overexpression of Fg-*usp7-ΔMATH* effectively rescued the attenuation of *ban*-lacZ induced by *usp7* knockdown. **g** Comparison of adult eye sizes from control flies (**g**-1), *usp7* knockdown (**g**-2), *usp7* overexpression (**g**-3), *usp7-CA* overexpression (**g**-4), and *usp7-ΔMATH* expression (**g**-5). Of note, knockdown of *usp7* decreased eye size, whereas overexpression of *usp7* or *usp7-ΔMATH* increased eye size. Overexpression of *usp7-CA* resulted in small eyes. Quantification analyses of the adult eyes were shown on right. **h** Comparison of adult eye sizes from control flies (**h**-1), *usp7* knockdown (**h**-2), *usp7* overexpression (**h**-3). Quantification analyses of the adult eyes were shown on right. Data are means ± SEM. $n = 10$ biological-independent eyes. $^{**}P < 0.01$ and $^{***}P < 0.001$ by Student's *t*-test. Scale bars: 50 μm for all wing discs (**a**-**f**), whereas 200 μm for all images (**g**, **h**)

(Fig. 7l). In sum, our results reveal that HAUSP positively regulates HCC tumorigenesis via stabilizing Yap, providing the HAUSP inhibitor as a potential drug for HCC clinical treatment.

## Discussion

The Hpo signaling has emerged as a critical and conserved mechanism that governs organ size and tumorigenesis[9]. Hpo pathway exerts its roles mainly through modulating the sub-cellular localization of the downstream co-transcriptional factor Yki[9,10]. Whether and how Hpo signaling regulates the abundance of Yki protein is still unknown. In this study, we demonstrated that the deubiquitinase Usp7 positively regulates Yki target gene

expression via deubiquitinating Yki (Fig. 8). We provided biochemical evidence that Usp7 bound Yki, which was inhibited by Hpo or Wts (Fig. 8). These results implicate that Hpo signaling not only prevents Yki nuclear accumulation, but also promotes nuclear Yki degradation via attenuating Yki-Usp7 interaction. In addition, the mammalian ortholog HAUSP played a conserved role in controlling Hpo pathway transduction. Finally, we revealed that the clinical HCC samples showed increased HAUSP protein and Yap protein, and that *hausp*-siRNA or HAUSP inhibitors treatment hampered proliferation of HCC cell lines. Our findings thus unveiled a conserved mechanism by which a DUB stabilizes Yki to achieve optimal Hpo pathway output, and

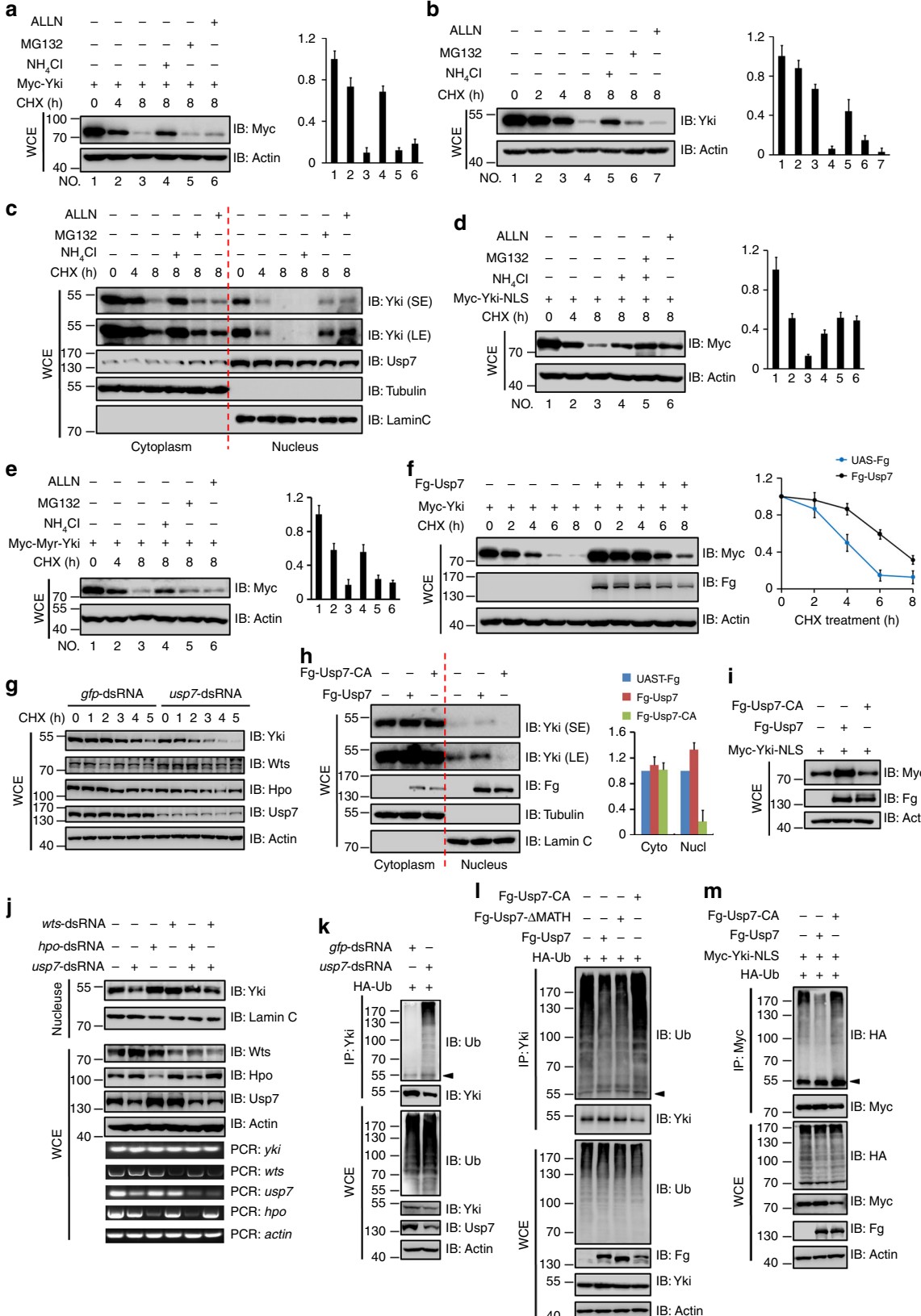

provide HAUSP as a potential drug target for Hpo-related cancer treatment.

In this study, we found that *usp7* steadily expressed throughout different developmental stages of *Drosophila* in a Hpo pathway-independent manner, indicating that Usp7 is likely a constitutive regulator of Yki without context specificity. Our data argue that

the homeostasis of Yki protein is governed by Hpo pathway activity. When Hpo signaling is closed, Usp7 stabilizes the nuclear Yki to turn on target genes expression (Fig. 8). Once Hpo pathway is activated, it not only inhibits Yki nuclear translocation via phosphorylating Yki, but also decreases nuclear Yki protein via attenuating Usp7-Yki affinity (Fig. 8). Therefore, Hpo pathway

**Fig. 5** Usp7 deubiquitinates and stabilizes Yorkie (Yki). **a** Immunoblots of lysates from S2 cells transfected with Myc-Yki and treated by CHX for indicated intervals. To prevent the function of lysosome or proteasome, the cells were treated with corresponding inhibitors. Of note, Myc-Yki was unstable, whereas lysosome inhibitor could hamper its degradation. Actin acts as a loading control. Quantification analyses were shown on right. **b** Endogenous Yki protein was unstable and underwent lysosome-mediated degradation. Actin acts as a loading control. Quantification analyses were shown on right. **c** S2 cells treated by CHX for indicated intervals were analyzed by subcellular fractionation. Of note, the cytoplasmic Yki was degraded by lysosome, whereas the nuclear Yki was degraded by proteasome. Lamin C and Tubulin are used as loading controls. **d** Myc-Yki-NLS protein was degraded by proteasome in S2 cells. Actin acts as a loading control. Quantification analyses were shown on right. **e** Myc-Myr-Yki protein was degraded by lysosome in S2 cells. Actin acts as a loading control. Quantification analyses were shown on right. **f** Immunoblots of lysates from S2 cells expressing indicated constructs and treated with CHX for indicated intervals. Quantification analyses were shown on right. Notably, Usp7 effectively blocked Yki degradation. **g** Knockdown of *usp7* promoted Yki degradation. **h** S2 cells transfected with indicated constructs were analyzed by subcellular fractionation. Of note, Usp7 increased the nuclear Yki protein levels. Quantification analyses were shown on right. Lamin C and Tubulin are used as loading controls. **i** Usp7 increased Myc-Yki-NLS protein level. **j** Knockdown of *usp7* decreased nuclear Yki protein. **k** Immunoblots of immunoprecipitates (top) or lysates (bottom three panels) from S2 cells expressing indicated proteins and treated with MG132 plus $NH_4Cl$ for 4 h. Knockdown of *usp7* promoted Yki ubiquitination. **l** Usp7 and Usp7-ΔMATH attenuated, but Usp7-CA promoted endogenous Yki ubiquitination. **m** Usp7 attenuated, whereas Usp7-CA promoted Yki-NLS ubiquitination. For statistical results, data are means ± SEM. $n = 3$ biological-independent experiments. Above all, the arrowhead indicates heavy IgG

weakens the function of Yki possibly through dual mechanisms (Fig. 8).

Although the previous findings have revealed that Yap undergoes β-TrCP-mediated ubiquitination and proteasome-dependent proteolysis[18], the Yki protein fails to be recognized by Slimb, the ortholog of β-TrCP in *Drosophila*. Our results implicated that the nuclear Yki was mainly degraded via proteasome, whereas the cytoplasmic Yki protein was destabilized by lysosome. These finding provided a possible explanation that the nuclear Yki terminates its signals via proteasome-mediated Yki destabilization. Consistently, the previous data have demonstrated that the cytoplasmic Yki binds and co-localizes with the arrestin domain-containing protein Leash, culminating in lysosome-mediated Yki degradation[57]. In mammalian cells, Yap is also targeted for lysosomal destabilization in some contexts[58,59]. Taken together, these results suggest that lysosome-dependent Yki/Yap proteolysis is likely a general mechanism.

The process of ubiquitination is triggered by the coordinated action of three classes of enzymes, including E1 ubiquitin-activating enzyme, E2 ubiquitin-conjugating enzyme, and E3 ubiquitin ligase[60]. In these enzymes, the E3 ligase determines the substrate specificity of the process. Although our studies have demonstrated that Yki undergoes ubiquitin-mediated proteolysis and Usp7 stabilizes Yki via deubiquitinating it, which E3 ligase accounts for Yki ubiquitination is still unknown. It will be fruitful to identify this E3 ligase.

Usp7 contains an N-terminally localized meprin and TRAF homology (MATH) domain, which is always responsible for protein interaction[61]. Previous studies have clearly demonstrated that Usp7 uses its MATH domain to bind several substrates, such as P53, Ci/Gli[31,32]. However, co-IP results in this study showed that the MATH domain of Usp7 was not essential for its interaction with Yki. In addition, overexpression of a truncated form Usp7-ΔMATH, in which the MATH domain is deleted, could effectively rescue the phenotypes caused by *usp7* knockdown. Our findings suggest that the MATH domain of Usp7 is dispensable for deubiquitinating some substrates. Although the C-terminal WW domains of Yki play important roles for it interacting with various partners, they were not involved in binding Usp7 because deletion of WW domains did not weaken its interaction with Usp7.

Increasing studies have implicated that HAUSP possibly plays contradictory roles in tumorigenesis. For instance, HAUSP stabilizes the P53 protein to act as a tumor suppressor[31]. However, our previous studies show that HAUSP promotes medulloblastoma cell survival and metastasis via deubiquitinating the Gli[54]. It is accepted that HAUSP plays an oncogenic role or a tumor-suppressor role possibly depends on the context. In this study, we found that HAUSP was upregulated in HCC samples and it positively regulated the proliferation of HCC cells via stabilizing Yap, suggesting that HAUSP acts as an oncogene in liver tumorigenesis. Consistently, a recent study revealed that HAUSP promoted HCC progression via deubiquitinating thyroid hormone receptor-interacting protein 12 (TRIP12)[52]. Through this study, we could cautiously add that, in addition to TRIP12, Yap was another target in HCC. Furthermore, HAUSP inhibitors apparently inhibited the proliferation of HCC cell lines, providing these inhibitors as potential drugs for HCC clinical treatment. It will be interesting to explore whether HAUSP inhibitors suppress liver tumorigenesis in vivo using mouse models.

## Methods

**DNA constructs**. To generate Myc-Yki, Myc-Mer, Myc-Ex, Myc-Kibra, Myc-Sav, Myc-Wts, Myc-Mats, Myc-14-3-3, HA-Hpo, HA-Wts, Myc-Sd, HA-Ub, and HA-Ci constructs, we amplified the corresponding complementary DNA (cDNA) fragments using Vazyme DNA polymerase (P505), and cloned them into the pUAST-Myc or pUAST-HA backbone vectors. HA-Hpo-KD, HA-Wts-KD constructs were kinase dead mutants[7,8]. Fg-Usp7 plasmid was a gift from Dr. Qing Zhang lab. Fg-Usp7-CA, Myc-Yki-SA, Myc-Yki-SD, Myc-Yki-KallR, Myc-Yki-K52R, Myc-Yki-K82R, Myc-Yki-K93R, Myc-Yki-K246R, and Myc-Yki-K274R were generated by PCR-based site-directed mutagenesis. Truncated constructs including Yki-N, Yki-C, Usp7-N, Usp7-M, Usp7-C, and Usp7-ΔMATH were generated by inserting the corresponding coding sequences into the backbone vectors. A nuclear localization signal (NLS) from SV40 (PPKKKRKV) was inserted at the C-terminus to generate Myc-Yki-NLS. A myristoylation signal (MGSSKSKPKDPSQRRRSLE) was inserted at the N-terminus to generate Myc-Myr-Yki. To generate Fg-HAUSP, Fg-HAUSP-CA, HA-β-TrCP, Myc-Yap, Myc-P53, Myc-Gli1, Myc-Gli2, and Myc-Gli3 for mammalian cell expression, we amplified these genes via cDNA pool from HCC samples and cloned them into CMV-Fg, pET-HA, or pcDNA3.1-Myc vectors. Myc-Yap-K90R was generated via PCR-based site-directed mutagenesis. Myc-LAST1, Myc-LAST2, Fg-MST1, and Fg-MST2 constructs have been described previously[21].

**Drosophila stocks**. Some stocks in this study are kindly provided from Dr. Qing Zhang lab[32], such as Fg-*usp7*, *usp7*[KG06814], and *usp7*-RNAi (#18231, VDRC). The mutant KG06814 (#14505, BDSC) is a P-element insertional allele of *usp7* (Fly-base). It is a lethal mutation caused by an insertion of P{SUPor-P} at the 5′-region of *usp7* gene. P{SUPor-P} was inserted in the opposite orientation relative to *usp7* transcription. En-gal4, Ay-gal4, ApG4, diap1-lacZ, ban-lacZ, *yki*-RNAi, *hpo*-RNAi, *wts*-RNAi, GMR-gal4, ptc-gal4, *usp7*-RNAi-2 (#34708, BDSC), and UAS-GFP were obtained from BDSC or VDRC. UAS-*wts*, UAS-*hpo*, and UAS-*yki* were from Dr. Jianhua Huang lab. UAS-*yki*-SA and *yki*[B5] had been described previously[8,9]. The Fg-*usp7*-CA, Fg-*usp7*-ΔMATH, Fg-*hausp*, and UAS-*yki*-1w transgenic flies were generated by injection of corresponding constructs into *Drosophila* embryos according to the standard methods[32]. To knockdown or overexpress genes, the male flies of RNAi or transgenes were crossed with indicated *gal4* virgin flies at 25 °C. All stocks used in this study were maintained and raised under standard conditions.

**Immunostaining**. Immunostaining of imaginal discs were performed with previous protocols[48]. In brief, third-instar larvae were dissected in PBS and fixed in freshly made 4% formaldehyde in phosphate buffered solution (PBS) at room temperature for 20 min, then washed three times with PBT (PBS plus 0.1% Triton X-100).

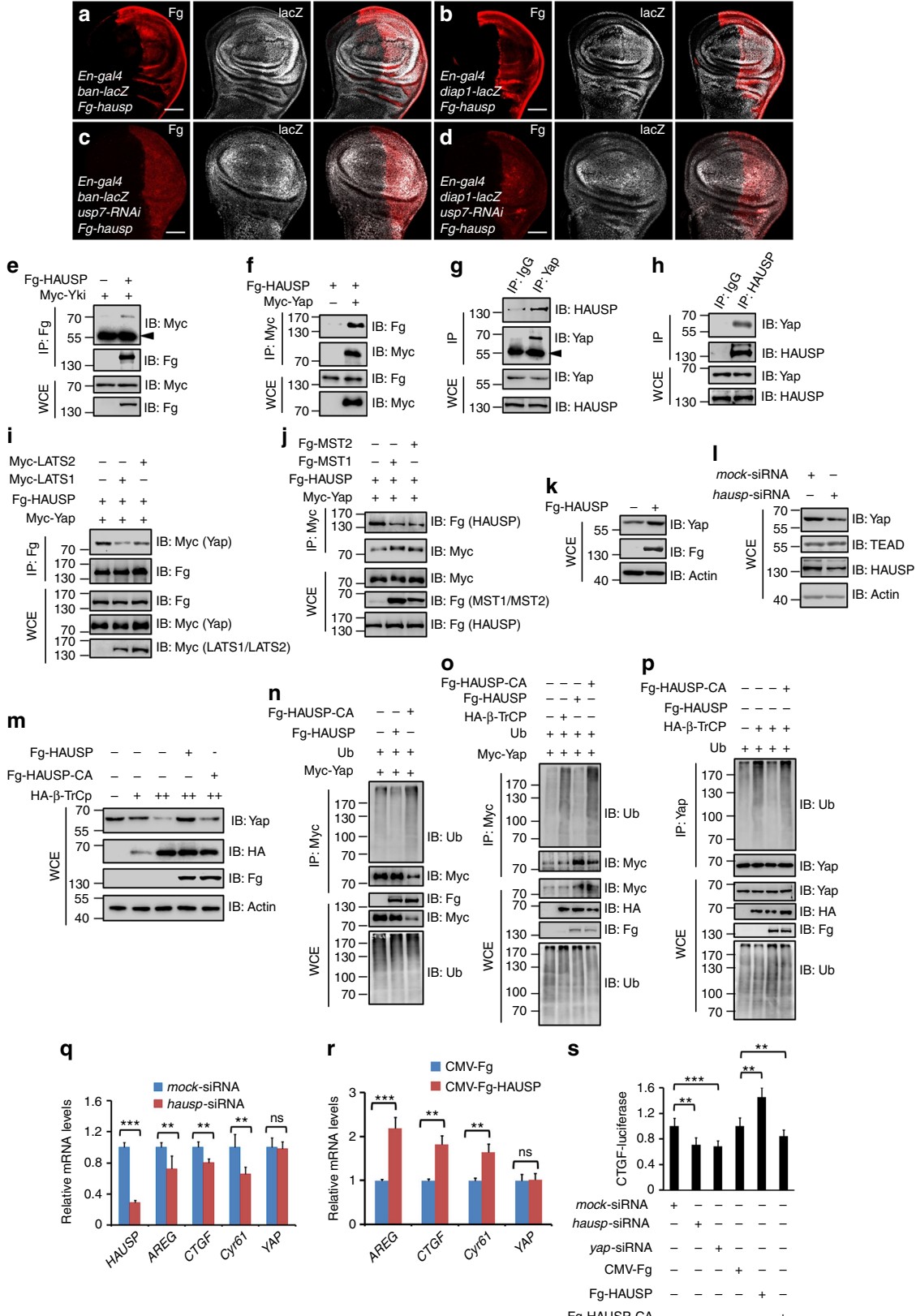

Larvae were incubated overnight with needed primary antibodies in PBT at 4 °C, then washed with PBT for three times and incubated with corresponding fluorophore-conjugated secondary antibody 2 h at room temperature. After washing for three times in PBT, discs were dissected and mounted in 40% glycerol. For immunostaining of S2 cells, cells were transfected with indicated constructs and stained with previous protocols[62]. Images were captured with Zeiss confocal microscope. Antibodies used in this study were as follows: rat anti-Ci (1:50; DSHB);

mouse anti-β Gal (1:500; Santa Cruz); goat anti-CycE (1:100; Santa Cruz); mouse anti-FgM2 (1:1000; Sigma); mouse anti-BrdU (1:10; DSHB); rabbit anti-Yki (1:1000), mouse anti-Myc (1:200; Santa Cruz), and DAPI (1:1000; Santa Cruz). Rabbit anti-DIAP1 antibody (1:100) was from Dr. Shigeo Hayashi lab. Mouse anti-Usp7 (1:100) was generated using aa1-234 as the antigen. Secondary antibodies used in this study were bought from Jackson ImmunoResearch, and were diluted at 1:500.

**Fig. 6** The regulation of Usp7 on Hippo (Hpo) pathway is conserved. **a**, **b** Wing discs expressing Fg-*hausp* by *En-gal4* were stained to show Fg (red) and *ban*-lacZ (white in **a**) or *diap1*-lacZ (white in **b**). All crosses were set at 29 °C. **c**, **d** Overexpression of *hausp* could restore the decrease of *ban*-lacZ (**c**) and *diap1*-lacZ (**d**) caused by *usp7* knockdown. Scale bars: 50 μm for all images. **e** Herpes virus-associated ubiquitin-specific protease (HAUSP) bound Yorkie (Yki) in S2 cells. The arrowhead indicates IgG. **f** HAUSP interacted with Yap. **g**, **h** Endogenous Yap reciprocally pulled down endogenous HAUSP in 293T cells. The cells from one 10 cm plate were lysed and the lysis was equivalently divided two parts for IP with control IgG or Yap/HAUSP antibodies. The expression of corresponding proteins from whole-cell lysis (WCE) was shown below. **i** LATS1/LATS2 decreased the interaction between HAUSP and Yap. **j** MST1/MST2 decreased the interaction between HAUSP and Yap. **k** Transfection of Fg-HAUSP increased endogenous Yap protein level in 293T cells. Actin acts as a loading control. **l** Knockdown of endogenous *hausp* decreased Yap protein, whereas did not affect TEAD level. Actin acts as a loading control. **m** Immunoblots of lysates from 293T cells transfected with indicated constructs. Notably, β-TrCP destabilized Yap in a dose-dependent manner, which was hampered by HAUSP, not HAUSP-CA. Actin acts as a loading control. **n** Immunoblots of immunoprecipitates (top) or lysates (bottom three panels) from 293T cells expressing indicated proteins and treated with MG132 plus NH$_4$Cl for 4 h. HAUSP decreased, whereas HAUSP-CA increased the ubiquitination of Yap protein. **o**, **p** HAUSP decreased, whereas HAUSP-CA increased β-TrCP-mediated Yap ubiquitination. **q** Relative mRNA levels of indicated genes from 293T cells with or without *hausp* knockdown were revealed by real-time PCR. **r** Relative mRNA levels of indicated genes from 293T cells with or without HAUSP transfection were revealed by real-time PCR. **s** CTGF-luciferase reporter assay in 293T cells transfected with indicated constructs or siRNAs. CTGF-luciferase activities were normalized to renilla luciferase activities. For statistical results, data are means ± SEM. $n = 3$ biological-independent experiments. In all above results, ${}^{**}P < 0.01$, ${}^{***}P < 0.001$, ns not significant, by Student's *t*-test

---

**Cell culture, transfection, immunoprecipitation, IB, luciferase reporter assay**. S2 cells (gift from Erjun Ling lab) were maintained in Schneider's *Drosophila* Medium (Gibco) with 10% heat-inactivated fetal bovine serum (FBS; Gibco) and 1% penicillin/streptomycin (Sangon Biotech). 293T and Huh7 cells were purchased from the ATCC and cultured in Dulbecco's modified Eagle's medium (Gibco) containing 10% FBS and 1% penicillin/streptomycin. In all, 7721 cells were cultured in 1640 medium (Gibco) containing 10% FBS and 1% penicillin/streptomycin. Cells were transfected using lipo2000 (Invitrogen) or polyethermide (Sigma) according to the manufacturer's instructions. Forty-eight hours after transfection, cells were harvested for immunoprecipitation (IP) and IB analysis with standard protocols. Uncropped versions of IBs are shown in Supplementary Figure 8. The following antibodies were used for IP and IB: mouse anti-HA (1:2000; Santa Cruz); mouse anti-Myc (1:2000 for IB, 1:200 for IP); rabbit anti-Myc (1:2000 for IB, 1:200 for IP; ABclonal); mouse anti-FgM2 (1:5000 for IB, 1:500 for IP); mouse anti-Ub (1:1000; Santa Cruz); mouse anti-Actin (1:5000; Genscript); mouse anti-Lamin C (1:1000; DSHB); mouse anti-β-Tubulin (1:1000; ABclonal); rabbit anti-Yki (1:10000); rabbit anti-Wts (1:5000); rabbit anti-Hpo (1:5000); mouse anti-Usp7 (1:2000 for IB, 1:200 for IP); mouse anti-Yap (1:1000 for IB, 1:100 for IP; Santa Cruz); rabbit anti-HAUSP (1:1000 for IB, 1:100 for IP; Santa Cruz); rabbit anti-TEAD (1:1000; Abcam); rabbit anti-AREG (1:1000; ABclonal); rabbit anti-Cyr61 (1:1000; ABclonal); rabbit anti-N-Myc (1:1000; ABclonal); rabbit anti-P53 (1:1000; ABclonal); rabbit anti-Gli1 (1:1000; ABclonal); rabbit anti-Gli2 (1:1000; ABclonal); rabbit anti-Gli3 (1:1000; ABclonal); goat anti-mouse HRP (1:10,000; Abmax), and goat anti-rabbit HRP (1:10,000; Abmax). Mouse anti-Yki (1:200 for IP) was generated using aa1-250 as the antigen. For dual luciferase reporter assay, 293T cells were transfected with the plasmids of CTGF-firefly luciferase and Pol III-Renilla luciferase. Twenty-four hours after transfection, cells were lysed in passive lysis buffer and luciferase activity was measured using a Dual Luciferase Assay Kit (Promega) according to manufacturer's instructions. For Usp7 inhibitors treatment, cells were treated by P5091 (#SML0070, Sigma), Usp7-IN (#HY-16709, MedChemExpress), or HBX-19818 (#HY-17540, MedChemExpress) at indicated concentrations for 24 h. All luciferase activity data are presented as means ± SEM of values from at least three experiments.

**Generating clones**. Clones were generated by FLP/FRT-mediated mitotic recombination[63]. Genotypes of the generated clones were as follows: *usp7* mutant clones in wing discs and eye discs: *usp7*$^{KG06814}$ FRT19A/hs-flp ubi-GFP FRT19A; control clones in wing discs and eye discs: FRT19A/hs-flp ubi-GFP FRT19A. The areas of clones and twin spots were measured by Image J. Ratios of clones to twin spots are presented as means ± SEM of values from at least 20 discs.

**GST fusion protein pull-down assay**. GST pull-down assays were carried out according to standard protocol[32]. Fusion proteins were induced by isopropyl β-D-thiogalactoside (IPTG) in *Escherichia coli* BL21, and purified with Beaver beads GSH (Beaverbio). GST fusion protein-loaded beads were incubated with cell lysates derived from S2 cells expressing corresponding proteins at 4 °C for 2 h. The beads were washed three times with PBS, followed by IB experiment.

**RNA interference**. To knockdown indicated genes in S2 cells, the double-stranded RNA (dsRNA) was generated by MEGAscript High Yield Transcription Kit (Ambion) according to the manufacturer's instructions. DNA templates targeting *usp7* (aa34-267 and aa981-1097), *wts* (aa1-177), and *hpo* (aa1-181) were generated by PCR and used for generating dsRNA. dsRNA targeting the *gfp* full-length coding sequence was used as a negative control. The indicated dsRNAs were transfected into S2 cells by ExFect 2000 (Vazyme) according to the manufacturer's instructions. To silence indicated genes in mammalian cells, siRNAs were transfected at a final concentration of 100 nM via lipo2000 according to the manufacturer's

instructions. For *hausp* knockdown, a mixture of two different siRNAs was used. The siRNA sequences were as follows: *mock*-siRNA: 5′-UUCUCCGAACGUGU CACGUdTdT-3′; *hausp*-siRNA-1: 5′-ACCCUUGGACAAUAUUCCUdTdT-3′; *hausp*-siRNA-2: 5′-AGUCGUUCAGUCGUCGUAUdTdT-3′; *yap*-siRNA: 5′-GACAUCUUCUGGUCAGAGAdTdT-3′; p53-siRNA-1: 5′-AAGGAAAUUUGC GUGUGGAdTdT-3′; p53-siRNA-2: 5′-GAUGAUUUGAUGCUGUCCCdTdT-3′; *gli1*-siRNA-1: 5′-AGCUUUCAUCAACUCGCGAdTdT-3′; *gli1*-siRNA-2: 5′-UCU GCCUAUACUGUCAGCCdTdT-3′; *gli2*-siRNA-1: 5′-GCGGUAGCUGCCCAAG GAGdTdT-3′; *gli2*-siRNA-2: 5′-AGCGGGUGCCCUCAGCCCAdTdT-3′; *gli3*-siRNA-1: 5′-AAAGUCCUGGACAGACUUAdTdT-3′; *gli3*-siRNA-2: 5′-GGGCUG AGCCCUACAGAUGdTdT-3′; N-Myc-siRNA-1: 5′-AAGAAGUUUGAGCUGC UGCdTdT-3′; N-Myc-siRNA-2: 5′-GCUGAUCCUCAAACGAUGCdTdT-3′.

**Protein stability assay and ubiquitination assay**. For protein stability assay, S2 cells were plated in 10-cm dishes and transfected with indicated plasmids. After 24 h, the cells were transferred into 12-well cell culture plates at equivalent densities. Cells were treated with 20 μg/ml CHX (Calbiochem) for the indicated intervals before harvesting. IB experiments were carried out to examine the levels of indicated proteins. In some experiments, MG132 (50 μM; Calbiochem) or ALLN (50 μM; Santa Cruz) was added to cultured cells to inhibit proteasome activity, whereas NH$_4$Cl (10 mM), leupeptin (500 μM; MedChenExpress), or chloroquine (15 μM; MedChenExpress) was used to block lysosome function. To separate the cytoplasmic and nuclear protein, we used nuclear and cytoplasmic protein extraction kit (#78833, Thermo Scientific) according to the manufacturer's instructions. For cell-based ubiquitination assays, S2 or 293T cells were transiently transfected with the indicated combinations of vectors. Four hours before cells harvesting, MG132 and NH$_4$Cl were added to the media to block proteasome-mediated and lysosome-mediated proteolysis. For cell-based ubiquitination assays, cells were lysed with denaturing buffer (1% SDS, 50 mM Tris-base, pH 7.5, 0.5 mM EDTA, and 1 mM DTT) and incubated at 100 °C for 5 min. The lysates were then diluted 10-fold with regular lysis buffer (50 mM Tris pH8.0, 0.1 M NaCl, 10 mM NaF, 1 mM Na$_3$VO$_4$, 0.5% NP-40, 10% Glycerol and 1 mM EDTA pH8.0) and subject to IP and IB analysis. In all IB experiments, the band intensity was measured by Image J. For quantitation, data are presented as means ± SEM of values from at least three independent experiments.

**RNA isolation, reverse transcription, and real-time PCR**. Cells or HCC samples were lysed in TRIzol (Invitrogen) for RNA isolation following standard protocols[32]. In all, 1 μg RNA was used for reverse transcription by HiScript® Q RT SuperMix with gDNA wiper (Vazyme) according to the instructions. Real-time PCR was performed on BIO-RAD CFX96™ with ChamQ SYBR® Color qPCR Master Mix (Q411, Vazyme). 2-ΔΔCt method was used for relative quantification. The primer pairs used were as follows: *hausp*, 5′-GGAAGCGGGAGATACAGATGA-3′ (forward) and 5′-AAGGACCGACTCACTCAGTCT-3′ (reverse); AREG, 5′-TCACT TTCCGTCTTGTTTTGG-3′ (forward) and 5′-CGGGAGCCGACTATGACTAC-3′ (reverse); CTGF, 5′-TAGGCTTGGAGATTTTGGGA-3′ (forward) and 5′-GGTTA CCAATGACAACGCCT-3′ (reverse); Cyr61, 5′-TATTCACAGGGTCTGCCCTC-3′ (forward) and 5′-AACGAGGACTGCAGCAAAA-3′ (reverse); AKD1, 5′-GTGT AGCACCAGATCCATCG-3′ (forward) and 5′-CGGTGAGACTGAACCGCTAT-3′ (reverse); *yap*, 5′-GCAACTCCAACCAGCAGCAACA-3′ (forward) and 5′-CG CAGCCTCTCCTTCTCCATCTG-3′ (reverse); *actin*, 5′-GATCATTGCTCCTCC TGAGC-3′ (forward) and 5′-ACTCCTGCTTGCTGATCCAC-3′ (reverse). Data are presented as means ± SEM of values from at least three experiments.

**MTT assay**. After indicated treatments, log-phase cells were seeded onto 96-well plates ($1 \times 10^3$–$10 \times 10^3$ cells per well). At 48 h after transfection, 10 μl MTT (5 mg/ml) was added, followed by additional incubation at 37 °C for 4 h before discarding

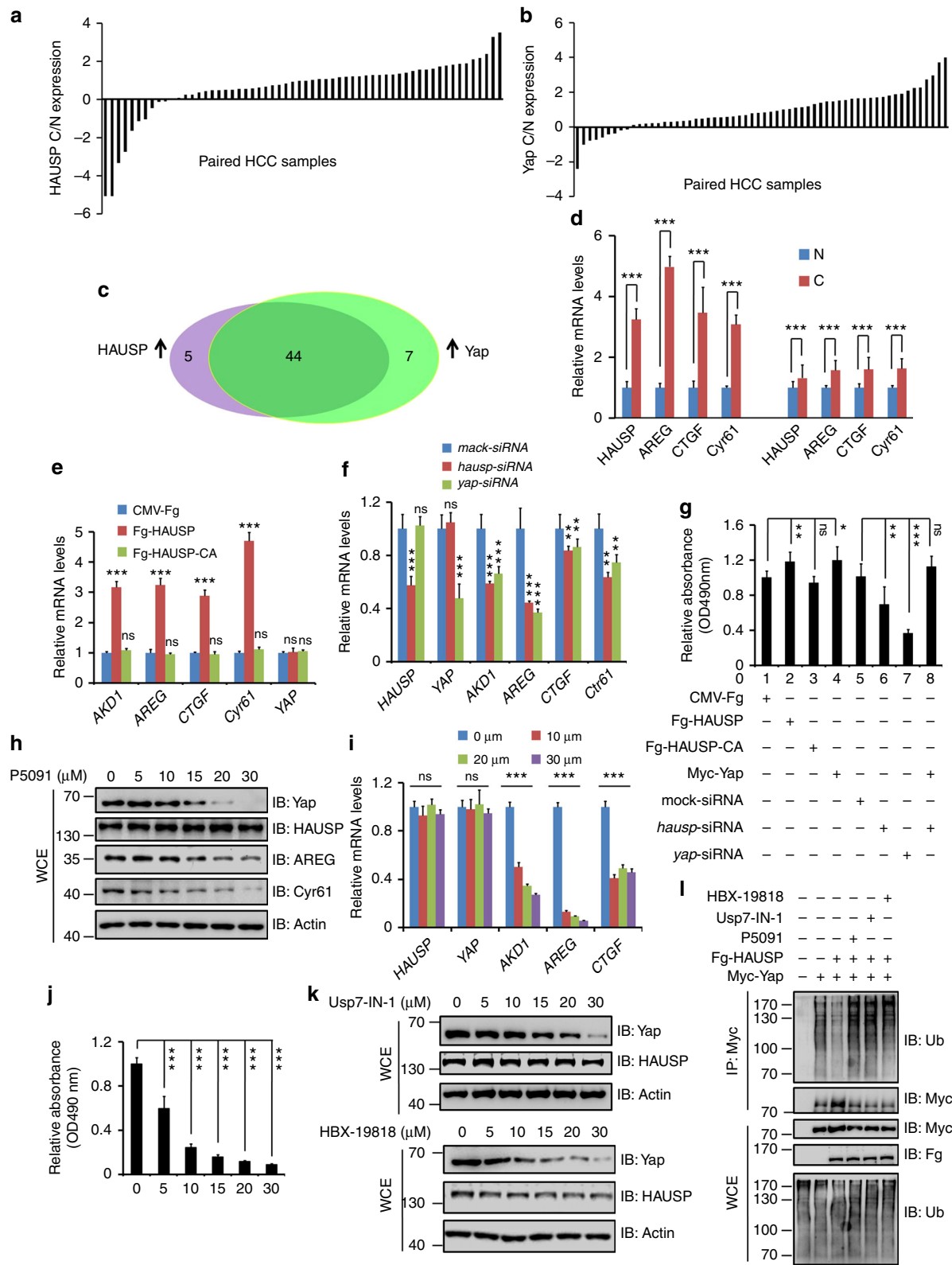

supernatants. Then, 100 μl dimethylsulfoxide was added to each well before shaking gently for 10 min to dissolve crystals. The absorbance of each well was measured at 490 nm using microplate reader. Data are presented as means ± SEM of values from at least three experiments.

**Patient samples.** Fresh-frozen primary HCC tissues and their paired normal samples were obtained from patients undergoing surgical resection at Zhuhai People's Hospital (Zhuhai, China) after consent was obtained from the patients.

None of the patients received any prior radiochemotherapy. For total protein extraction, place the equal amount tissues (40 mg) in tubes and grind the tissue with a plastic rod for 50–60 times with twisting force on ice. Then, add five times cell lysis buffer (50 mM Tris pH 8.0, 0.1 M NaCl, 10 mM NaF, 1 mM Na$_3$VO$_4$, 0.5% NP-40, 10% Glycerol and 1 mM EDTA pH8.0) and continue to grind for 50–60 times. Cap the tube and incubate on ice for 10–15 min. Centrifuge at 12,000 rpm for 15 min. The supernatant was subject to IB assay following standard protocols.

**Fig. 7** Herpes virus-associated ubiquitin-specific protease (HAUSP) and Yap are upregulated in hepatocellular carcinoma (HCC) samples. **a**, **b** The relative expression of HAUSP (**a**) and Yap (**b**) in 60 pairs of HCC and normal liver samples. Each bar was the $\log_2$ value of the ratio of HAUSP or Yap protein levels between HCC (C) and matched normal tissues (N) from the same patient. The bar > 0 represented HAUSP was upregulated in HCC sample because $\log_2 1 = 0$, vice versa. **c** Comparison between HAUSP elevation and Yap elevation in HCC samples. The overlapping region represented both HAUSP and Yap elevation cases. **d** Relative mRNA levels of indicated genes from HCC samples and matched normal tissues were revealed by real-time PCR. **e** Relative mRNA levels of indicated genes from Huh7 cells were revealed by real-time PCR. **f** Relative mRNA levels of indicated genes from Huh7 cells were revealed by real-time PCR. **g** MTT results of Huh7 cells transfected with indicated constructs or siRNAs. **h** Immunoblots of lysates from Huh7 cells treated with Usp7 inhibitor P5091 at indicated concentrations for 24 h. Actin acts as a loading control. **i** Relative mRNA levels of indicated genes from Huh7 cells treated with P5091 at distinct concentrations for 24 h. **j** Cell proliferation of Huh7 cells treated with P5091 at indicated concentrations for 24 h. **k** Immunoblots of lysates from Huh7 cells treated with Usp7-IN-1 or HBX-19818 at indicated concentrations for 24 h. Actin acts as a loading control. **l** Usp7 inhibitors suppressed Yap ubiquitination. Quantative data are shown as means ± SEM. $n = 3$ biological-independent experiments. In all above results, $^{**}P < 0.01$, $^{***}P < 0.001$, ns, not significant by Student's $t$-test

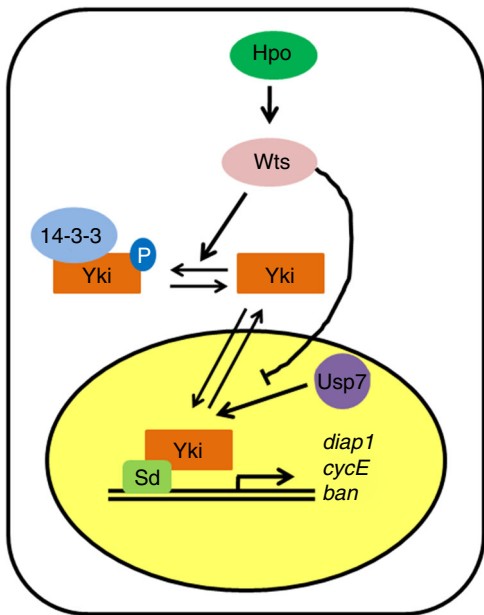

**Fig. 8** A proposed model of Usp7 deubiquitinating Yorkie (Yki). When Hippo (Hpo) pathway turns on, Hpo phosphorylates and activates Wts. In turn, Wts phosphorylates the transcriptional coactivator Yki, culminating Yki retention in the cytoplasm via binding with 14-3-3. When Hpo pathway is closed, Yki translocates into the nucleus to activate the target gene expression with the assistance of Sd. The nuclear Yki protein is destabilized by proteasome, which is attenuated by Usp7

**Statistical analysis**. The density of IB band was measured by Image J software. Statistical analysis was performed with GraphPad Prism software. The data shown in the figures were representative of three or more independent experiments and were analyzed by one way Student's $t$-test, and $P < 0.05$ was considered statistically significant. Where exact $P$-values are not shown, statistical significance is shown as with $^{*}P < 0.05$, $^{**}P < 0.01$, and $^{***}P < 0.001$.

## Data availability
All relevant data are available from the corresponding author upon reasonable request.

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

## Acknowledgements

We sincerely thank Dr. Qing Zhang (Nanjing University, China) for generous providing some stocks and plasmids. We also thank Dr. Erjun Ling (Shanghai Institute for Biological Sciences, China), Dr. Jianhua Huang (Zhejiang University, China), and Dr. Shigeo Hayashi (RIKEN Brain Science Institute, Japan) for providing reagents. We also appreciate National Institute of Genetics of Japan (NIG), Bloomington Stock Center (BSC), and Developmental Studies Hybridoma Bank at the University of Iowa for providing fly stocks and reagents. We thank Dr. Susumu Hirose (National Institute of Genetics, Japan) and Yasushi Hiromi (National Institute of Genetics, Japan) for discussions and comments on the manuscript. This work was supported by grants from the Shandong Agricultural University talent fund (72119), the Natural Science Foundation of Shandong Province (ZR2017MC014), Funds of "Shandong Double Tops" Program (SYL2017YSTD09), the National Natural Science Foundation of China (31802012, 31602011, 31571502, 31471319), the National Key Research and Development Program of China (2017YFA0205200), and grant from the Construction Engineering Special Fund of "Taishan Scholars" (no. ts201712022).

## Author contributions

The author(s) have made the following declarations about their contributions: Z.Z., Q.L., and L.L. designed the experiments. X.S., Y.D., M.Z., Yan Li, D.G., G.W., Y.G., and Yong Li performed the experiments. L.L. and M.Z. provided the patient samples. Z.Z. and S.W. carried out data analysis. Z.Z. wrote the manuscript with the help of all authors.
