## [Peer Review File · Nature Communications]

Reviewers' comments:

Reviewer #1 (Remarks to the Author):

In this manuscript, the authors describe USP7/HAUSP as a negative regulator of Yki/YAP, the main transcription factor of the Hippo pathway. Hippo pathway is critical in organ development and tissue homeostasis. A defective Hippo pathway leads to the activation of the oncogene Yki/Yap, to its nuclear translocation and to the transcription of cell proliferation and survival genes, favouring the development of cancer. Deubiquitylases (DUBs) remove the ubiquitin moiety from substrates thus rescuing substrates from degradation. DUBs are druggable proteins that are of interest in cancer research due to their role in the stabilisation of oncogenes. There is a limited number of DUBs that have been associated to the Hippo pathway so far.

The authors describe the modulation of the expression of Yki target genes such as *ban*, *diap1* and Cyclin E upon USP7 depletion, using hypomorphic allele of USP7 or overexpression of USP7 in the eye and wing imaginal discs of *Drosophila*. The authors can rescue the decrease of *ban* and *diap1* triggered by USP7 depletion by the overexpression of USP7. The authors performed co-immunoprecipitation to map the interaction of USP7 with Yki using overexpressed truncated proteins. The authors show that the MATH domain of USP7 and the WW domain of Yki, 2 domains involved in protein-protein interaction, are dispensable for USP7-Yki interaction. The authors performed series of experiments to assess the role of USP7 in the stabilisation of nuclear Yki. Finally the authors describe the correlation between elevated USP7 protein levels and Yki protein levels in hepatocarcinoma cells. The authors used a USP7 inhibitor to test the effect of USP7 inhibition on the expression of *Yap*, Yki ortholog, and *Yap* target genes.

The role of a USP7 in Hippo pathway and especially in the stabilisation of *Yap* is novel. This discovery would be important for the clinical positioning of the current USP7 inhibitors. However I think some key experiments are missing.

The authors should

- 1) assess that USP7 acts on Hippo signalling independently of its known role on other transcription factors such as p53, Gli and MYCN, using genetic tools and/or inhibitors,
- 2) map the domains of USP7 and *Yap* that are involved in their interaction,
- 3) identify the lysines of *Yap* that are deubiquitylated by USP7,
- 4) check that USP7 specifically deubiquitylates nuclear, active *Yap* and not the inactive, phosphorylated, cytoplasmic *Yap*, using phosphomimetic and phosphorylation-deficient *Yap* mutants.

Minor points

Related to the introduction:

There are more recent reviews on DUBs than Wilkinson, 2000. Although there are still no review describing the 7 DUBs families, and not 5 as stated line 80, the authors should refer to a more recent review on DUBs (Mevisen and Komander, 2017, *Annuals Reviews of Biochemistry*) and to the articles describing the 2 new families, MINDY and ZUFSP (Kwasna et al., 2018, *Molecular Cell*; Haar et al., 2018, *Molecular Cell*; Harmanns et al., 2018, *Nature communications*; Rehman et al., 2016, *Molecular Cell*).

In Figure 1a-d, the authors assess the levels of *diap1*, *ban* and Cyclin E upon USP7 depletion in the imaginal wing disc. Supplementary Figure 1g show the levels of USP7 depletion using USP7-RNAi under *En* promoter. The authors should move the Sup Fig 1g that shows USP7 depletion as control to the main figure.

Line 146 "overexpression of *usp7* substantially increased the levels of *diap1-lacZ*, *CycE* and *ban-lacZ*" is misleading in the absence of quantitation. Quantification of the decrease of USP7 protein levels, of *diap1-lacZ*, *ban-lacZ* and *CycE* upon USP7 depletion is missing. In general, the authors should add quantitation when they claim that there is a decrease or rescue of a phenotype. Details on how USP7 knockdown is performed and general info on the flies are missing in the

Methods.

Line 105-106: The sentence "To remove the possibility that USP7 specifically modulate diap1 transcription" is misleading. USP7 could regulate the transcription of several genes. To check for the absence of transcription regulation, the authors need to perform QPCR in *usp7*-RNAi context and assess USP7 mRNA levels, Yki mRNA levels, *ban* mRNA levels, *diap1* mRNA levels and *CycE* mRNA levels. Only partial data are shown Supplementary Figure 2e.

Related to IP blots: Please label the blots representing the IP, and the WCL for Figure 3c, 3d, 3g, 3h, 3i, 3k, 3l, 6e, 6f, 6i. For all IPs blots, please show the proteins levels of both proteins, the target protein and the one used for the pull down, in the WCL and in the IP. Please show the full blots for Figure 3i, to assess the lack of interaction between USP7 and Yki-C. The authors should also perform the Myc IP.

The authors showed that USP7 and USP7-MATH can both pull down Yki, which is in contradiction with the sentence line 204 that states "USP7 binds Yki in a MATH-dependent manner". Please correct the sentence.

Related to lysosomal degradation inhibition (Figure 5a-d): NH_4Cl is not a specific inhibitor of lysosomal degradation. The authors should use a specific lysosomal degradation inhibitor such as leupeptin.

In Figure 5c, the author should show the half-life of Myc-Yki-NLS compared to endogenous nuclear Yki. The authors could perform fractionation cellular and cytoplasmic fractionation to see how the cytoplasmic fraction of Yki and the nuclear fraction respond to proteasome or lysosomal inhibitors.

Related to western-blot and IP blots: please add a loading control for all cell lysates blots. Please specify the lysis buffer used. Please add the protein ladders to all blots.

In Figure 5e, it is not clear from the legend and methods if the proteins are from total lysates or fractionation.

The authors claim that 1) Yki is mainly cytoplasmic (supplementary figure 5), 2) the degradation of Yki is mainly through the lysosome (Figure 1a-c), 3) the effect of USP7 is on the nuclear fraction of Yki (Figure 5f). It is then surprising to see an effect on total Yki when USP7 is depleted as shown in Figure 5e.

Moreover, the depletion of USP7 is very inefficient. The authors should perform the knockdown with several independent USP7 oligos.

Length of USP7 knockdown is not indicated in the Methods.

In Figure 5e, USP7 depletion decreases the levels of Yki. The authors should give more details on Figure 5e about how the half-life of Yki was determined. The authors should explain why Yki protein levels are similar in USP7 depleted cells and non-depleted cells at the start of the cycloheximide treatment.

In Figure 5f, the authors should add the quantitation of Yki. The authors should show blots with lower exposure to help the reader to see the absence of change of cytoplasmic Yki protein levels. The authors should add the western-blot of endogenous USP7, overexpressed USP7 WT and overexpressed USP7 CA in the fractionation shown Figure 5f.

The authors should specify how they separated the cytoplasmic and nuclear proteins in the Methods. The sentence line 529 "To separate the cytoplasmic and nuclear protein, we used nuclear and cytoplasmic protein extraction kit according to the manufacturer's instruction" is not giving enough information to the readers to be able to repeat the experiment.

Line 537-538 the authors mention "lysates were then diluted 10-fold with regular lysis buffer". They should give more details to the reader about what the 'regular lysis buffer' is.

Contrary that what has been shown Figure 3g, there is a decrease of Yki with overexpression of

USP7-MATH in the whole cellular lysates in the Figure 5j. The authors should add a loading control.

If the claim is that USP7 is changing the ubiquitylation of Yki in the nucleus, the authors should perform the ubiquitylation assay after fractionation, and show both cytoplasmic and nuclear fraction.

Related to the QPCR experiments: Please add the mRNA levels for Yap1 for Figures 6q, 6r, 7e and 7f, and the mRNA levels of HAUSP for Figure 7f.

In Figure 6s, the authors should use an ANOVA and not a t-test.

Related to Figure 7 h-j, a recent publication describes the interaction of USP10, USP47 and OTULIND1 with Yap, but not USP7 (Yao et al., 2018, Nat. comms). OTULIND1 is shown to deubiquitylate Yap, but not USP10 and USP47. Could the authors discuss about the discrepancy of the interaction of USP7 with Yap.

In light of this paper, since P5091 is also inhibiting USP47 (Weinstock et al., 2012, ACS Med Chem Lett), The authors should use a specific USP7 inhibitor that is not inhibiting USP10 and/or USP47 (Turnbull et al., 2017, Nature).

There is no information on how the authors extracted the proteins from the HCC samples in the Methods.

Reviewer #2 (Remarks to the Author):

In their manuscript "Usp7 regulates Hippo pathway through deubiquitinating the transcriptional coactivator Yorkie," Sun et al identified the deubiquitinase Usp7 as a positive regulator of Hippo signaling in *Drosophila*. Mechanistically, Usp7 binds to Yki, blocks its ubiquitination and degradation, and thus stabilizes Yki's protein level and promotes the expression of its target genes. The Hippo pathway inhibits the physical association of Usp7-Yki, and negatively regulates Yki protein stability. Importantly, they showed that these functions of Usp7 were conserved by its human ortholog HAUSP, and reported a positive correlation of YAP and HAUSP expression in HCC samples. Overall, the findings are intriguing, the data are solid and the manuscript is well written. I think that it will be of significant interest to the readers of Nature Communication. Yet, I do have concerns with some aspects of the manuscript.

1. The molecular nature of the *usp7*(KG06814) allele should be described. In addition, since this allele has a P-element inserted in the 5' UTR, which presumably affect the expression of *usp7*. This should be examined by Usp7 antibody in the mutant clones.
2. The authors should pay attention to the figure magnification, for example, wing discs in Fig. 11-1n should be adjusted to the same magnification, and scale bars should be added into the panels.
3. For supplemental Fig. 2f & 2g, overexpression of Usp7 alone should be added as a control.
4. In line 204, "...in a MATH-dependent manner" should be "MATH-independent manner".
5. Does in vivo expression of Hpo-KD and/or Wts-KD increase Yki level? If yes, could loss of *usp7* suppress this enhancement?
6. In line 222, "Supplementary Fig. 3b" should be "Supplementary Fig. 4b".
7. For Fig. 4g-k, it is not clear why UAS-wts is included? Actually UAS-wts is not necessary here. A better genetics essay would be measuring the eye sizes of GMR-Gal4 control, GMR-Gal4 driven expression of *usp7* RNAi, or *usp7* and its variants in wild-type (+/+) and heterozygous *yki* (+/*yki*-) background.
8. In Fig. 5e, gfp-dsRNA is labelled in the wrong lanes, should be in the first 4 lanes!
9. Does HAUSP expression rescue the lethality of *usp7*(KG06814) mutants? And the wing phenotype of Ap-Gal4 UAS-*usp7*-RNAi (Fig. S1f)?

Reviewers' comments:**Reviewer #1 (Remarks to the Author):**

In this manuscript, the authors describe USP7/HAUSP as a negative regulator of Yki/YAP, the main transcription factor of the Hippo pathway. Hippo pathway is critical in organ development and tissue homeostasis. A defective Hippo pathway leads to the activation of the oncogene Yki/Yap, to its nuclear translocation and to the transcription of cell proliferation and survival genes, favouring the development of cancer. Deubiquitylases (DUBs) remove the ubiquitin moiety from substrates thus rescuing substrates from degradation. DUBs are druggable proteins that are of interest in cancer research due to their role in the stabilisation of oncogenes. There is a limited number of DUBs that have been associated to the Hippo pathway so far.

The authors describe the modulation of the expression of Yki target genes such as ban, diap1 and Cyclin E upon USP7 depletion, using hypomorphic allele of USP7 or overexpression of USP7 in the eye and wing imaginal discs of *Drosophila*. The authors can rescue the decrease of ban and diap1 triggered by USP7 depletion by the overexpression of USP7. The authors performed co-immunoprecipitation to map the interaction of USP7 with Yki using overexpressed truncated proteins. The authors show that the MATH domain of USP7 and the WW domain of Yki, 2 domains involved in protein-protein interaction, are dispensable for USP7-Yki interaction. The authors performed series of experiments to assess the role of USP7 in the stabilisation

of nuclear Yki. Finally the authors describe the correlation between elevated USP7 protein levels and Yki protein levels in hepatocarcinoma cells. The authors used a USP7 inhibitor to test the effect of USP7 inhibition on the expression of Yap, Yki ortholog, and Yap target genes.

The role of a USP7 in Hippo pathway and especially in the stabilisation of Yap is novel. This discovery would be important for the clinical positioning of the current USP7 inhibitors. However I think some key experiments are missing.

The authors should

1) *assess that USP7 acts on Hippo signalling independently of its known role on other transcription factors such as p53, Gli and MYCN, using genetic tools and/or inhibitors.*

Response: Thank you! It is a good suggestion. The transcription factors P53, Gli and N-Myc are well-known Usp7/HAUSP targets. To remove the possibility that Usp7/HAUSP upregulates Yki/Yap indirectly through these transcription factors, we carry out the following experiments. First, the RT-qPCR results show that either *usp7/hausp* knockdown or *usp7/hausp* overexpression regulates Hippo pathway outputs, but does not affect *yki/yap* mRNA levels (Fig. 6q-r and Supplementary Fig. 2e), suggesting that Usp7/HAUSP regulates Yki/Yap not at transcriptional level. Second, in the wing disc, Ci (the ortholog of Gli) only expresses in the anterior (A) compartment. Knockdown of *usp7* in the wing disc via *ApG4*, which expresses across

the A and P compartments, equally decreased A- and P-compartmental *diap1-lacZ* levels (Supplementary Fig. 1f), removing the possibility that Usp7 modulates Hippo pathway through Ci. Third, knockdown of *p53*, *gli1*, *gli2*, *gli3* or *N-Myc* does not decrease Yap protein levels, while *hausp* knockdown decreases Yap protein (Supplementary Fig. 6b). Fourth, overexpression of P53, Gli1, Gli2 or Gli3 does not increase Yap protein, whereas HAUSP could elevate Yap (Supplementary Fig. 6c-d). Fifth, knockdown of *p53* (#45138 from VDRC) in *Drosophila* wing discs does not affect the expression of Hippo pathway target genes (Figure below). Taken together, these results suggest that Usp7/HAUSP acts on Hippo pathway independent of P53, Ci/Gli and N-Myc.

2) map the domains of USP7 and Yap that are involved in their interaction.

Response: Thanks a lot! Through co-IP experiments, we find that Yki binds Usp7 via its N-terminal domain (Fig. 3i-j). To map which domain of Usp7 is responsible for its

binding to Yki, we generated various truncated constructs. The co-IP assays show that Usp7 binds Yki-N through its C-terminal domain (Supplementary Fig. 3j).

3) *identify the lysines of Yap that are deubiquitylated by USP7.*

Response: Thank you very much! It is really a good question. Known to us, ubiquitin is always attached to the lysine residue of the substrate. To examine whether the ubiquitin modification of Yki occurs on lysine, we first replace all 14 lysines by arginines to generate Myc-Yki-KallR mutant. Yki-KallR is stable and fails to be ubiquitination (Supplementary Fig. 5d, e), supporting that ubiquitins attach to lysine residues on Yki protein. To further identify which lysines are responsible for ubiquitination, we carry out a prediction via an online web server (<http://bdmpub.biocuckoo.org>) and identify five potential ubiquitination sites (K52, K82, K93, K246 and K274). Compared with wild type Yki, only mutation of K93 (K93R) significantly blocks Yki degradation (Supplementary Fig. 5f-g). In line with this, K to R mutation on 93 apparently attenuates Yki ubiquitination (Supplementary Fig. 5h), together supporting that K93 of Yki is the ubiquitination site. Intriguingly, this ubiquitination site is evolutionarily conserved between *Drosophila* and human (Supplementary Fig. 6e). Mutation of the corresponding lysine (K90) to arginine blocks Yap degradation (Supplementary Fig. 6f) and ubiquitination (Supplementary Fig. 6g). Meanwhile, we find that Yap-K90R shows higher efficiency in promoting Huh7 cell proliferation than wild type Yap (Supplementary Fig. 6h), together

suggesting that K90 is a potential ubiquitination site of Yap protein. Taken together, our results have demonstrated that the ubiquitination site is K93 on Yki or K90 on Yap.

4) check that USP7 specifically deubiquitylates nuclear, active Yap and not the inactive, phosphorylated, cytoplasmic Yap, using phosphomimetic and phosphorylation-deficient Yap mutants.

Response: Thanks! It is a good suggestion. Through immunostaining, we find that Usp7 mainly localizes in the nucleus, suggesting that Usp7 possibly plays roles in the nucleus. Indeed, we separate the cytoplasmic and nuclear protein for ubiquitination assay and find that Usp7 mainly deubiquitinates the nuclear Yki (Supplementary Fig. 5a). We also employ two Yki mutants, Yki-NLS and Myr-Yki, which exclusively localizes in the nucleus and cytoplasm respectively, to validate our results. As expected, Usp7 shows high efficiency to deubiquitinating Yki-NLS (Supplementary Fig. 5b). In addition, we find that Usp7 shows higher activity to deubiquitinating phosphorylation-deficient Yki mutant (Yki-SA) than phosphomimetic Yki mutant (Yki-SD) (Supplementary Fig. 5c). Taken together, we provide enough evidence to support that Usp7 mainly deubiquitinates nuclear Yki. However, we also observe that a small amount of Usp7 protein localizes in the cytoplasm (Fig. 5c). Thus, Usp7 also shows weak activity to deubiquitinating Myr-Yki and Yki-SD (Supplementary Fig. 5b-c).

Minor points

Related to the introduction:

*There are more recent reviews on DUBs than Wilkinson, 2000. Although there are still no review describing the 7 DUBs families, and not 5 as stated line 80, the authors should refer to a more recent review on DUBs (Mevisen and Komander, 2017, *Annals Reviews of Biochemistry*) and to the articles describing the 2 new families, MINDY and ZUFSP (Kwasna et al., 2018, *Molecular Cell*; Haar et al., 2018, *Molecular Cell*; Harmanns et al., 2018, *Nature communications*; Rehman et al., 2016, *Molecular Cell*).*

Response: Thank you! I apologize for not following the recent papers. In this revised manuscript, I make modification according to your suggestion.

In Figure 1a-d, the authors assess the levels of diap1, ban and Cyclin E upon USP7 depletion in the imaginal wing disc. Supplementary Figure 1g shows the levels of USP7 depletion using USP7-RNAi under En promoter. The authors should move the Sup Fig 1g that shows USP7 depletion as control to the main figure.

Response: Thank you! It is a good suggestion. I have adjusted the indicated figure according to your suggestion.

Line 146 "overexpression of usp7 substantially increased the levels of diap1-lacz, CycE and ban-lacz" is misleading in the absence of quantitation. Quantification of the decrease of USP7 protein levels, of diap1-lacz, ban-lacz and CycE upon USP7 depletion is missing. In general, the authors should add quantitation when they claim that there is a decrease or rescue of a phenotype.

Response: Thanks! I have added the quantitation in this revised manuscript (**Fig. 1q**).

Details on how USP7 knockdown is performed and general info on the flies are missing in the Methods.

Response: Thank you! I have added the stock number of *usp7* RNAi lines in this revised manuscript. The readers could search the detailed information of these flies via on line website (<http://flybase.org>). In addition, I have added the method of how to silence or overexpress a gene in *Drosophila*.

Line 105-106: The sentence "To remove the possibility that USP7 specifically modulate diap1 transcription" is misleading. USP7 could regulate the transcription of several genes. To check for the absence of transcription regulation, the authors need to perform QPCR in usp7-RNAi context and assess USP7 mRNA levels, Yki mRNA levels, ban mRNA levels, diap1 mRNA levels and CycE mRNA levels. Only partial

data are shown Supplementary Figure 2e.

Response: Thank you very much! I have supplemented the RT-PCR results to Supplementary Fig 2e. I am sorry that the sentence is misleading. Thus, I modified this sentence in this revised manuscript. The readout *diap1-lacZ* (or *ban-lacZ*) is a fly stock which the lacZ coding sequence inserts downstream of *diap1* (or *ban*) promoter. Thus, *diap1-lacZ* (or *ban-lacZ*) monitors the transcription level of *diap1* (or *ban*). In our manuscript, we find that knockdown of *usp7* decreased *diap1-lacZ* and *ban-lacZ*, suggesting that *usp7* knockdown suppresses the transcription of *diap1* and *ban*. In fact, *ban* encodes a microRNA, which expression is activated by Yki. It is hard to examine the mRNA level of *ban* through RT-PCR. *ban-lacZ* provides a useful tool to monitor the transcription of *ban* gene.

Related to IP blots: Please label the blots representing the IP, and the WCL for Figure 3c, 3d, 3g, 3h, 3i, 3k, 3l, 6e, 6f, 6i. For all IPs blots, please show the proteins levels of both proteins, the target protein and the one used for the pull down, in the WCL and in the IP. Please show the full blots for Figure 3i, to assess the lack of interaction between USP7 and Yki-C. The authors should also perform the Myc IP.

Response: Thank you! I have added all IP blots and made modification in this revised manuscript.

The authors showed that USP7 and USP7-MATH can both pull down Yki, which is in contradiction with the sentence line 204 that states "USP7 binds Yki in a MATH-dependent manner". Please correct the sentence.

Response: Thank you for reading our manuscript carefully. I sincerely apologize for this mistake. I have corrected this mistake in this revised manuscript.

Related to lysosomal degradation inhibition (Figure 5a-d): NH₄Cl is not a specific inhibitor of lysosomal degradation. The authors should use a specific lysosomal degradation inhibitor such as leupeptin.

Response: Thank you for your constructive suggestion. I have employed two additional lysosome inhibitors, leupeptin and chloroquine. Either leupeptin or chloroquine treatment get the same result as NH₄Cl (Supplementary Fig. 4j-k).

In Figure 5c, the author should show the half-life of Myc-Yki-NLS compared to endogenous nuclear Yki. The authors could perform fractionation cellular and cytoplasmic fractionation to see how the cytoplasmic fraction of Yki and the nuclear fraction respond to proteasome or lysosomal inhibitors.

Response: Thank you! S2 cell treated with CHX plus different inhibitors for indicated intervals are underwent cell fractionation to get cytoplasmic protein and nuclear

protein. Immunoblots show that the cytoplasmic Yki is degraded by lysosome, while the nuclear Yki is destabilized by proteasome (Fig. 5c). In addition, we employ Yki-NLS and Myr-Yki variants and find that Yki-NLS is degraded by proteasome (Fig. 5d and Supplementary Fig. 4j), whereas Myr-Yki is degraded by lysosome (Fig. 5e and Supplementary Fig. 4k). Taken together, the cytoplasmic Yki and nuclear Yki undergo degradation via distinct mechanisms.

Related to western-blots and IP blots: please add a loading control for all cell lysates blots. Please specify the lysis buffer used. Please add the protein ladders to all blots.

Response: Thank you for your suggestion. I have added loading control and ladders in this revised manuscript. I have provided detailed information of cell lysis buffer.

In Figure 5e, it is not clear from the legend and methods if the proteins are from total lysates or fractionation.

Response: I am sorry. I have added markers in the figure. The proteins are from total lysates (Fig. 5g in the revised manuscript).

The authors claim that 1) Yki is mainly cytoplasmic (supplementary figure 5), 2) the degradation of Yki is mainly through the lysosome (Figure 1a-c), 3) the effect of USP7 is on the nuclear fraction of Yki (Figure 5f). It is then surprising to see an effect on

total Yki when USP7 is depleted as shown in Figure 5e.

Response: Thank you! It is a good question. First, accumulating studies have shown that Yki protein shuttles between the cytoplasm and nucleus. Yki protein keeps basal amount in the nucleus. Knockdown of *usp7* mainly promotes nuclear Yki degradation. Meanwhile, the cytoplasmic Yki will translocate into the nucleus for degradation. Thus, knockdown of *usp7* decreases total Yki. Second, Usp7 indeed mainly localizes in the nucleus, but a small amount of Usp7 protein stays in the cytoplasm (Fig. 5c). I sincerely hope these explanations are satisfactory.

Moreover, the depletion of USP7 is very inefficient. The authors should perform the knockdown with several independent USP7 oligos.

Response: Thanks! I have repeated this experiment according to your suggestion and get a better result (Fig. 5g).

Length of USP7 knockdown is not indicated in the Methods.

Response: I have added the detailed information of *usp7* dsRNA in this revised manuscript.

In Figure 5e, USP7 depletion decreases the levels of Yki. The authors should give

more details on Figure 5e about how the half-life of Yki was determined. The authors should explain why Yki protein levels are similar in USP7 depleted cells and non-depleted cells at the start of the cycloheximide treatment.

Response: Thank you for your suggestion. I have repeated this experiment using mixed dsRNAs to improve efficiency of *usp7* knockdown. As shown in Fig. 5g, knockdown of *usp7* not only decreases Yki protein, but also promotes Yki degradation. According to this result, the half-life of Yki is about 4hrs without *usp7* dsRNA. Knockdown of *usp7* will shorten the half-life of Yki to about 2hrs.

In Figure 5f, the authors should add the quantitation of Yki. The authors should show blots with lower exposure to help the reader to see the absence of change of cytoplasmic Yki protein levels. The authors should add the western-blot of endogenous USP7, overexpressed USP7 WT and overexpressed USP7 CA in the fractionation shown Figure 5f.

Response: Thank you! I have added quantitation in this revised manuscript and added a lower exposure figure (Fig. 5h).

The authors should specify how they separated the cytoplasmic and nuclear proteins in the Methods. The sentence line 529 "To separate the cytoplasmic and nuclear protein, we used nuclear and cytoplasmic protein extraction kit according to the

manufacturer's instruction" is not giving enough information to the readers to be able to repeat the experiment.

Response: Thank you! I have added the detailed information about the kit in this revised manuscript.

Line 537-538 the authors mention "lysates were then diluted 10-fold with regular lysis buffer". They should give more details to the reader about what the 'regular lysis buffer' is.

Response: Thank you! I have added the detailed information about regular cell lysis buffer in this revised manuscript.

Contrary that what has been shown Figure 3g, there is a decrease of Yki with overexpression of USP7-MATH in the whole cellular lysates in the Figure 5j. The authors should add a loading control.

Response: Thank you! I have rerun the sample and added the loading control (Fig. 5l).

If the claim is that USP7 is changing the ubiquitylation of Yki in the nucleus, the authors should perform the ubiquitylation assay after fractionation, and show both

cytoplasmic and nuclear fraction.

Response: Thank you for your constructive suggestion. I have performed this experiment according to your suggestion. We separate the cytoplasmic and nuclear protein for ubiquitination assay and find that Usp7 mainly deubiquitinates the nuclear Yki (Supplementary Fig. 5a).

Related to the QPCR experiments: Please add the mRNA levels for Yap1 for Figures 6q, 6r, 7e and 7f, and the mRNA levels of HAUSP for Figure 7f.

Response: I have added the results according to your suggestion.

In Figure 6s, the authors should use an ANOVA and not a t-test.

Response: I am sorry for misleading. In Fig. 6s, we only compare two groups at one time (*mock*-siRNA compared with *hausp*-siRNA, *mock*-siRNA compared with *yap*-siRNA, CMV-Fg compared with Fg-HAUSP, or CMV-Fg compared with Fg-HAUSP-CA). Therefore, t-test is more suitable than ANOVA. To avoid the misleading, I have adjusted the figure.

Related to Figure 7 h-j, a recent publication describes the interaction of USP10, USP47 and OTULIND1 with Yap, but not USP7 (Yao et al., 2018, Nat. comms).

OTULIND1 is shown to deubiquitylate Yap, but not USP10 and USP47. Could the authors discuss about the discrepancy of the interaction of USP7 with Yap.

Response: It is a good question. As a matter of fact, I have read this paper before I submitted this manuscript. I think this discrepancy is possible due to distinct experimental conditions. I have noted that the cell lysis buffers are different in two studies. The formula of cell lysis buffer in our manuscript is: 50mM Tris pH8.0, 0.1M NaCl, 10mM NaF, 1mM Na₃VO₄, 0.5% NP-40, 10% Glycerol and 1mM EDTA pH8.0. While the formula in Yao paper is: 120mM NaCl, 20mM NaF, 1mM EDTA, 25mM Tris pH7.5 and 0.33% CHAPS. As shown, the ion concentration in our reagent is lower. In addition, the detergents used in two studies are distinct. These differences possibly bring some distinct results. In our manuscript, we validate Yki-Usp7 interaction using co-IP and GST pull-down assays.

In light of this paper, since P5091 is also inhibiting USP47 (Weinstock et al., 2012, ACS Med Chem Lett), The authors should use a specific USP7 inhibitor that is not inhibiting USP10 and/or USP47 (Turnbull et al., 2017, Nature).

Response: Thank you for your suggestion. I have employed two other specific inhibitors, Usp7-IN-1 (Kessler, 2014) and HBX-19818 (Reverdy et al., 2012) to confirm our results. As shown, either Usp7-IN-1 or HBX-19818 treatment decreases Yap protein in a concentration-dependent manner, without affecting HAUSP levels

(Fig. 7k). In addition, Usp7-IN-1 and HBX-19818 inhibit the proliferation of 7721 cells and Huh7 cells (Supplementary Fig. 7d). Consistently, we also find that Usp7-IN-1 and HBX-19818 could effectively block HAUSP-mediated Yap deubiquitination (Fig. 7l). In wing discs, knockdown of *usp10* (#37859 from VDRC) or *usp47* (#26027 from VDRC) does not affect *ban-lacZ* expression (Figures below), suggesting that Usp10 and Usp47 are not involved in Hippo pathway regulation. In 293T cells, knockdown of *usp10* or *usp47* does not decrease Yap protein (Figures below), indicating that Usp10 and Usp47 do not stabilize Yap.

There is no information on how the authors extracted the proteins from the HCC samples in the Methods.

Response: Thank you for your suggestion. I have added the detailed protocols to

extract total protein from HCC samples in this revised manuscript.

Reviewer #2 (Remarks to the Author):

In their manuscript "Usp7 regulates Hippo pathway through deubiquitinating the transcriptional coactivator Yorkie," Sun et al identified the deubiquitinase Usp7 as a positive regulator of Hippo signaling in *Drosophila*. Mechanistically, Usp7 binds to Yki, blocks its ubiquitination and degradation, and thus stabilizes Yki's protein level and promotes the expression of its target genes. The Hippo pathway inhibits the physical association of Usp7-Yki, and negatively regulates Yki protein stability. Importantly, they showed that these functions of Usp7 were conserved by its human ortholog HAUSP, and reported a positive correlation of YAP and HAUSP expression in HCC samples. Overall, the findings are intriguing, the data are solid and the manuscript is well written. I think that it will be of significant interest to the readers of Nature Communication. Yet, I do have concerns with some aspects of the manuscript.

1. The molecular nature of the usp7(KG06814) allele should be described. In addition, since this allele has a P-element inserted in the 5' UTR, which presumably affect the expression of usp7. This should be examined by Usp7 antibody in the mutant clones.

Response: Thank you for your constructive suggestion. I have added the detailed information of *usp7*^{KG06814} mutant allele in this revised manuscript. Due to the

homozygote of $usp7^{KG06814}$ is lethal before the first instar larvae, we balance $usp7^{KG06814}$ with FM7. *Kr*-GFP and select GFP negative embryos for immunoblotting. As shown in Supplementary Fig. 1g, $usp7^{KG06814}$ homozygotes have no Usp7 expression, indicating that $usp7^{KG06814}$ is a null allele. As a matter of fact, the previous study has clearly demonstrated that this mutant is a null allele (van der Knaap et al., 2005).

2. The authors should pay attention to the figure magnification, for example, wing discs in Fig. 1l-ln should be adjusted to the same magnification, and scale bars should be added into the panels.

Response: Thank you very much! I have adjusted the magnification and added scale bars in this revised manuscript according to your suggestion.

3. For supplemental Fig. 2f & 2g, overexpression of Usp7 alone should be added as a control.

Response: Thanks! I have added the control disc in this revised manuscript according to your suggestion.

4. In line 204, "...in a MATH-dependent manner" should be "MATH-independent manner".

Response: Thank you for reading our manuscript carefully. I sincerely apologize for this mistake. I have corrected this mistake in this revised manuscript.

5. *Does in vivo expression of Hpo-KD and/or Wts-KD increase Yki level? If yes, could loss of usp7 suppress this enhancement?*

Response: It is really a good suggestion. As expected, knockdown of *hpo* or *wts* elevated Yki, which was restored by *usp7* RNAi (Supplementary Fig. 3I), suggesting that Hpo/Wts downregulates Yki, at least in part, through Usp7.

6. *In line 222, “Supplementary Fig. 3b” should be “Supplementary Fig. 4b”.*

Response: Thank you for reading our manuscript carefully. I sincerely apologize for this mistake. I have corrected this mistake in this revised manuscript.

7. *For Fig. 4g-k, it is not clear why UAS-wts is included? Actually UAS-wts is not necessary here. A better genetics essay would be measuring the eye sizes of GMR-Gal4 control, GMR-Gal4 driven expression of usp7 RNAi, or usp7 and its variants in wild-type (+/+) and heterozygous yki (+/yki-) background.*

Response: It is a good question. The immunostaining results have clearly

demonstrated that Usp7 is a negative regulator of Hippo pathway. To address whether Usp7 regulates eye sizes through Hippo pathway, we chose a Hippo-activated background. This background will improve the sensitivity for these assays. In addition, knockdown of *usp7* indeed decreases eye size under heterozygous *yki* background, while overexpression of *usp7* exerts an opposite role (Fig. 4h).

8. In Fig. 5e, *gfp-dsRNA* is labelled in the wrong lanes, should be in the first 4 lanes!

Response: I am sorry for this mistake. I have corrected this mistake in this revised manuscript.

9. Does HAUSP expression rescue the lethality of *usp7*(KG06814) mutants? And the wing phenotype of *Ap-Gal4 UAS-usp7-RNAi* (Fig. S1f)?

Response: Good questions! We have performed the rescue assay and found that overexpression of *hausp* indeed restores the small wing caused by *usp7* RNAi (Supplementary Fig. 1a). In addition, overexpression of *hausp* using *actin-gal4* could rescue the lethality of *usp7*^{KG06814} mutants (Supplementary Fig. 6a). Taken together, our data strong support that HAUSP could replace Usp7 in *Drosophila*.

References

Kessler, B.M. (2014). Selective and reversible inhibitors of ubiquitin-specific protease 7: a patent evaluation (WO2013030218). Expert opinion on therapeutic patents 24, 597-602.

Reverdy, C., Conrath, S., Lopez, R., Planquette, C., Atmanene, C., Collura, V., Harpon, J., Battaglia, V., Vivat, V., Sippl, W., *et al.* (2012). Discovery of specific inhibitors of human USP7/HAUSP deubiquitinating enzyme. *Chemistry & biology* *19*, 467-477.

van der Knaap, J.A., Kumar, B.R., Moshkin, Y.M., Langenberg, K., Krijgsveld, J., Heck, A.J., Karch, F., and Verrijzer, C.P. (2005). GMP synthetase stimulates histone H2B deubiquitylation by the epigenetic silencer USP7. *Molecular cell* *17*, 695-707.

REVIEWERS' COMMENTS:

Reviewer #1 (Remarks to the Author):

The authors have satisfactorily addressed my previous comments in the resubmitted manuscript. I am pleased to recommend publication in the present form.

Reviewer #2 (Remarks to the Author):

All my concerns have been satisfactorily addressed.